# Variance-quantitative trait loci enable systematic discovery of gene-environment interactions for cardiometabolic serum biomarkers

Kenneth E. Westerman [1,2,3✉], Timothy D. Majarian[2], Franco Giulianini[4], Dong-Keun Jang[2], Jenkai Miao[5], Jose C. Florez [2,3,6], Han Chen[7,8], Daniel I. Chasman [4,9,10,11], Miriam S. Udler [2,3,6], Alisa K. Manning[1,2,3] & Joanne B. Cole [2,5,6✉]

Gene-environment interactions represent the modification of genetic effects by environmental exposures and are critical for understanding disease and informing personalized medicine. These often induce differential phenotypic variance across genotypes; these variance-quantitative trait loci can be prioritized in a two-stage interaction detection strategy to greatly reduce the computational and statistical burden and enable testing of a broader range of exposures. We perform genome-wide variance-quantitative trait locus analysis for 20 serum cardiometabolic biomarkers by multi-ancestry meta-analysis of 350,016 unrelated participants in the UK Biobank, identifying 182 independent locus-biomarker pairs ($p < 4.5 \times 10^{-9}$). Most are concentrated in a small subset (4%) of loci with genome-wide significant main effects, and 44% replicate ($p < 0.05$) in the Women's Genome Health Study ($N = 23,294$). Next, we test each locus-biomarker pair for interaction across 2380 exposures, identifying 847 significant interactions ($p < 2.4 \times 10^{-7}$), of which 132 are independent ($p < 0.05$) after accounting for correlation between exposures. Specific examples demonstrate interaction of triglyceride-associated variants with distinct body mass- versus body fat-related exposures as well as genotype-specific associations between alcohol consumption and liver stress at the *ADH1B* gene. Our catalog of variance-quantitative trait loci and gene-environment interactions is publicly available in an online portal.

[1] Clinical and Translational Epidemiology Unit, Mongan Institute, Massachusetts General Hospital, Boston, MA, USA. [2] Programs in Metabolism and Medical & Population Genetics, Broad Institute of Harvard and MIT, Cambridge, MA, USA. [3] Department of Medicine, Harvard Medical School, Boston, MA, USA. [4] Division of Preventive Medicine, Brigham and Women's Hospital, Boston, MA, USA. [5] Division of Endocrinology, Boston Children's Hospital, Boston, MA, USA. [6] Diabetes Unit and Center for Genomic Medicine, Massachusetts General Hospital, Boston, MA, USA. [7] Human Genetics Center, Department of Epidemiology, Human Genetics and Environmental Sciences, School of Public Health, The University of Texas Health Science Center at Houston, Houston, TX, USA. [8] Center for Precision Health, School of Biomedical Informatics, The University of Texas Health Science Center at Houston, Houston, TX, USA. [9] Division of Genetics, Brigham and Women's Hospital, Boston, MA 02115, USA. [10] Medical and Population Genetics Program, Broad Institute, Cambridge, MA, USA. [11] Department of Epidemiology, Harvard T.H. Chan School of Public Health, Boston, MA, USA. ✉email: kewesterman@mgh.harvard.edu; jcole@broadinstitute.org

D espite advances in identifying the genetic and environmental determinants of common complex diseases like cardiovascular disease and type 2 diabetes, the variability in the penetrance of genetic effects and the role played by environmental factors across populations are not fully understood. Part of this variability is due to gene-environment interactions (GEIs), in which genetic and non-genetic exposures synergistically affect disease-related traits. Understanding how exposures, including demographic, physiological, and lifestyle, modify genetic effects can spur new biological insights and therapies. Conversely, clarifying the ways in which one's genetic background alters the effect of environmental exposures is a key step toward genome-guided precision medicine. There has been a substantial amount of hypothesis-driven research into GEIs for cardiometabolic traits[1], though these interactions often lack substantial replication[2,3]. Genome-wide interaction studies have begun to more systematically explore these interactions across the genome[4], and across multiple exposures[5], but have focused primarily on body mass index (BMI).

Comprehensive mapping of cardiometabolic GEIs across all genome-wide genetic variants and possible exposures carries practical, computational, and statistical challenges. Practically, it is difficult to collect and examine the thousands of exposures necessary for an "exposome-wide" approach, though recent software tools have made the process of high-dimensional phenotype processing substantially easier[6]. Meanwhile, the massive number of GEI tests involved renders such an endeavor computationally infeasible and statistically underpowered, compounding known power limitations due to typically modest GEI effect sizes and low breadth and precision of exposure measurements[7]. Many screening procedures have been proposed to circumvent these computational and statistical limitations by reducing the genetic search space. These strategies prioritize specific sets of variants for GEI testing, such as those with main effects on the outcome[8] or the exposure[9].

GEIs may induce differences in the variance of continuous phenotypes across genotypes. Thus, tests for genetic markers associated with this variance, termed variance-quantitative trait loci (vQTLs), represent an alternate strategy to identify loci harboring underlying GEIs for quantitative traits[10–16]. vQTLs can be identified in genome-wide scans analogous to those testing for phenotypic mean differences in typical genome-wide association studies (GWAS). Though direct GEI tests are more powerful when the environmental factor is measured accurately, vQTL tests may be advantageous when the relevant exposure is unknown or poorly measured, or when multiple exposures have aligned GEI effects at a locus. Such scenarios are quite common in practice due to the high dimensionality, dense correlation structure, and poor measurement of typical environmental exposures[17]. Recent studies in the UK Biobank (UKB) have demonstrated that a genome-wide vQTL discovery approach for anthropometric and lung function-related traits prioritizes variants enriched for GEI effects[15,16]. However, it is unclear whether vQTL effects are observed more broadly for cardiometabolic traits and to what degree the specific underlying GEI relationships can be uncovered using a more comprehensive array of environmental exposures.

Our objective was to identify genetic variants associated with the variance of a series of cardiometabolic blood biomarkers and to leverage these associations to efficiently detect underlying GEIs at an exposome-wide scale. We conducted a multi-ancestry, genome-wide vQTL scan to prioritize genetic loci across 20 biomarkers in UKB ($N = 350,016$) in stage one, followed by an exposome-wide interaction study (EWIS), incorporating 2380 exposures, to identify the specific underlying GEIs in stage two. We found 136 vQTLs, many of which were pleiotropic across

biomarkers and largely overlapped with known GWAS loci. Using our EWIS approach, we then identified >800 specific GEIs with numerous correlated exposures underlying 43 of our vQTLs. Our study develops novel genetic maps of variance effects on a panel of cardiometabolic biomarkers, greatly increases the breadth of exposures tested for GEI, and introduces a publicly available catalog of vQTLs and GEIs that can inform precision medicine related to cardiometabolic health.

## Results

**vQTLs are common and concentrated in known GWAS loci**. The primary analysis consisted of two stages: first, the identification of vQTLs via genome-wide analysis using Levene's test[18], and second, the exploration of underlying GEIs at these loci across thousands of exposures (workflow described in Fig. 1). Each stage was conducted in each of four ancestry groups in the population-based UKB cohort (Supplementary Data 1), though the European-ancestry subset was by far the largest, comprising 96% of the sample. Twenty cardiometabolic serum biomarkers were examined in this study, including lipids, lipoproteins, glycemic traits, liver enzymes, and kidney function markers (biomarkers listed in Supplementary Data 2; pre-processed biomarker distributions shown in Supp. Fig. S1).

In stage one, a study-wide Bonferroni significance threshold of $p < 4.51 \times 10^{-9}$ was established to correct for testing of 11.1 effective biomarkers (see Methods). The meta-analysis identified 184 vQTL-biomarker pairs at 136 independent loci after distance-based pruning (Fig. 2a, Supplementary Data 3). While most vQTLs were biomarker-specific, a modest proportion was "pleiotropic" with respect to phenotypic variance, with five loci common to at least four biomarkers (Fig. 2b). The locus surrounding rs5117 near the *APOE/APOC* cluster was the most pleiotropic, associated with the variance of nine biomarkers (including lipids [TC, LDL-C, HDL-C, TG], lipoproteins [ApoA, ApoB, LipA], liver enzymes [ALT], and hsCRP). This locus has strong main effects on the same biomarkers and is known to interact with lifestyle factors in determining cardiovascular disease risk[19].

P-values from the vQTL meta-analysis tracked closely with those from the European subset (Supp. Fig. S2), while uncovering ten additional loci that were not significant in the European subset alone. Six of these reflected contributions from multiple ancestries, while four were variants that only reached sufficient frequency in one ancestry. In contrast, nine vQTL-biomarker pairs reached significance in the European-only analysis but not in the meta-analysis. There were also 71 ancestry-specific vQTLs reaching significance in one or more ancestry-specific analyses but not the meta-analysis (Supplementary Data 4), 61 of which were found in non-European-ancestry groups. While these ancestry-specific vQTLs may indicate the presence of heterogeneous variance effects across populations due to genetic ancestry differences or ethnic differences in environmental exposures, it is also possible that the non-European findings are the result of spurious associations given lower sample sizes, especially at lower minor allele frequencies[15]. Therefore, downstream analysis focuses on the meta-analysis vQTL findings.

To understand these vQTLs in the context of genetic main effects (MEs), standard GWAS were also conducted for the same adjusted biomarker phenotypes. The resulting genetic MEs showed strong overlap with the identified vQTLs (Fig. 2c, Supplementary Data 5). The majority of vQTLs had significant main effects: across all biomarkers, 90.5% of vQTLs were in ME loci. However, the converse was not true: only 3.7% of ME loci contained vQTLs. Thus, while this analysis did not discover many novel loci, it substantially narrowed the genomic search space and

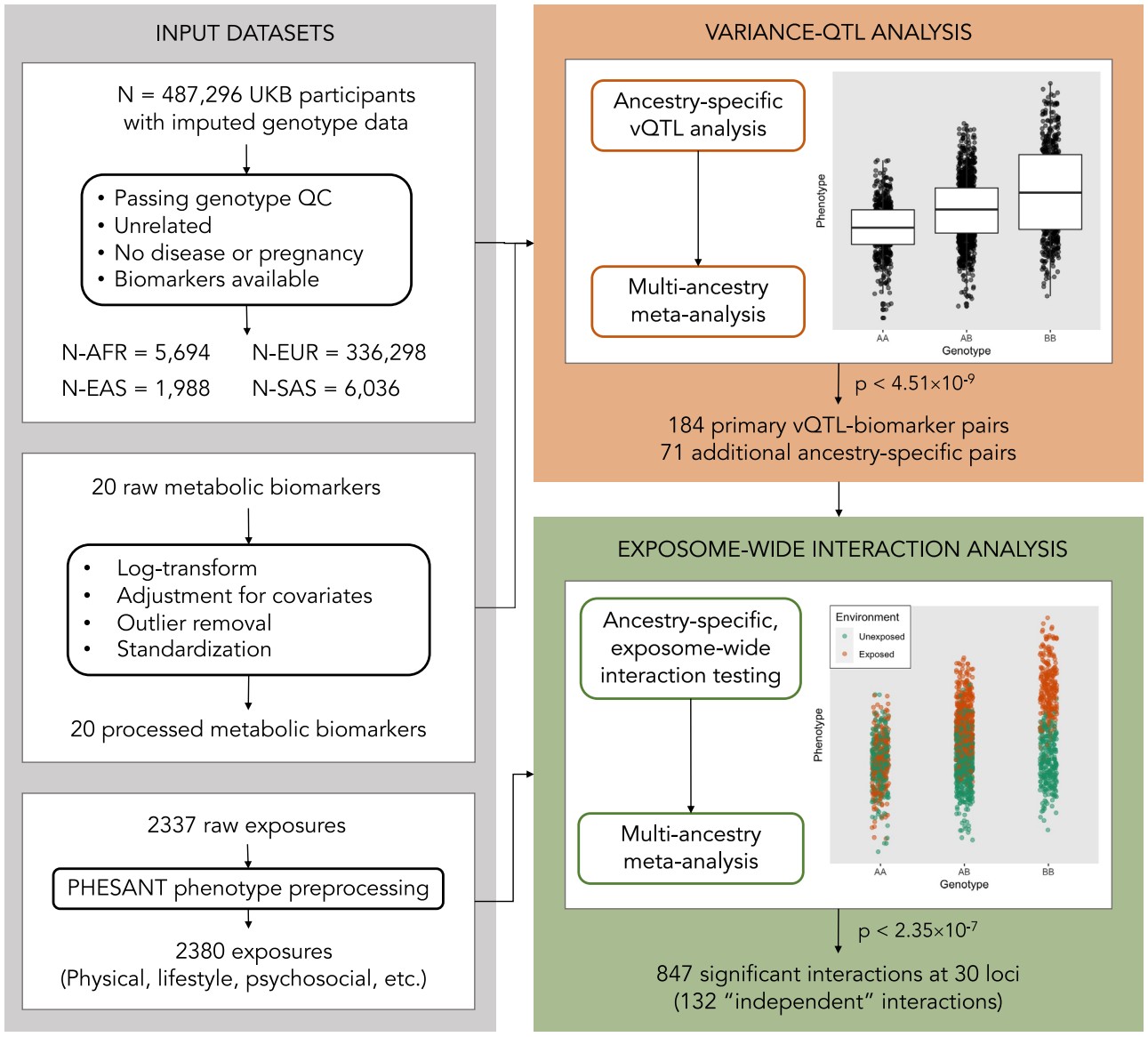

**Fig. 1 Analysis workflow.** Unrelated individuals without major disease from any of four ancestry groups in the UK Biobank were included in the analysis. Twenty metabolic biomarkers were pre-processed, including log transformation, adjustment for biological and technical covariates, and outlier removal. vQTL analysis was performed genome-wide, separately in each ancestry group followed by a multi-ancestry meta-analysis. Significant variant-biomarker pairs were taken forward for GEI analysis, along with 2380 exposures pre-processed using the PHESANT tool. Finally, GEI tests were performed for each combination of variant-biomarker-exposure triplet, again using ancestry-stratified analysis followed by meta-analysis.

therefore multiple testing burden for downstream analysis as compared to starting with the set of loci from a standard mean-effect GWAS. Creatinine was particularly notable in this comparison, having the greatest number of MEs but no vQTLs – this could be explained by a true lack of underlying GEIs or gene-gene interactions, or by the limited power of vQTL approaches to detect more-complex interactions (e.g., involving multiple exposures in opposite directions). In contrast, Lipoprotein A was especially enriched for vQTLs, with 6 of its 9 MEs having vQTL associations. Beyond simple overlap of loci, we observed a quantitative relationship between vQTL and ME significances, which persisted when examining specific biomarkers whose raw values were either normally distributed (HbA1c) or non-normally distributed (GGT) (Supp. Fig. S3). This quantitative relationship confirms a similar result found for BMI in the UKB, and may be due to the fact that interactions are more likely to be present at loci with established biological

connections to the phenotype of interest[16,20]. vQTL and ME loci did not show significant differential enrichment for epigenomic features using the Locus Overlap Analysis (LOLA) method[21] (see Methods).

The 17 vQTL-biomarker pairs without corresponding main effects deserve specific consideration. First, we note that they had a wide distribution of p-values (including eight with $p > 0.05$) rather than clustering just above the Bonferroni significance threshold. Thus, some of these loci may truly lack genetic main effects, especially given the greater power of standard linear regression compared to Levene's test[11]. Upon deeper investigation, we note that these 17 vQTLs were evenly spread across both chromosomes and biomarkers. While there were more vQTLs with at least one GEI among those that also had a ME (28%) compared to a vQTL effect only (18%), this difference was not significant (Chi-square test $p = 0.55$) and does not provide evidence that vQTLs without a ME represent spurious

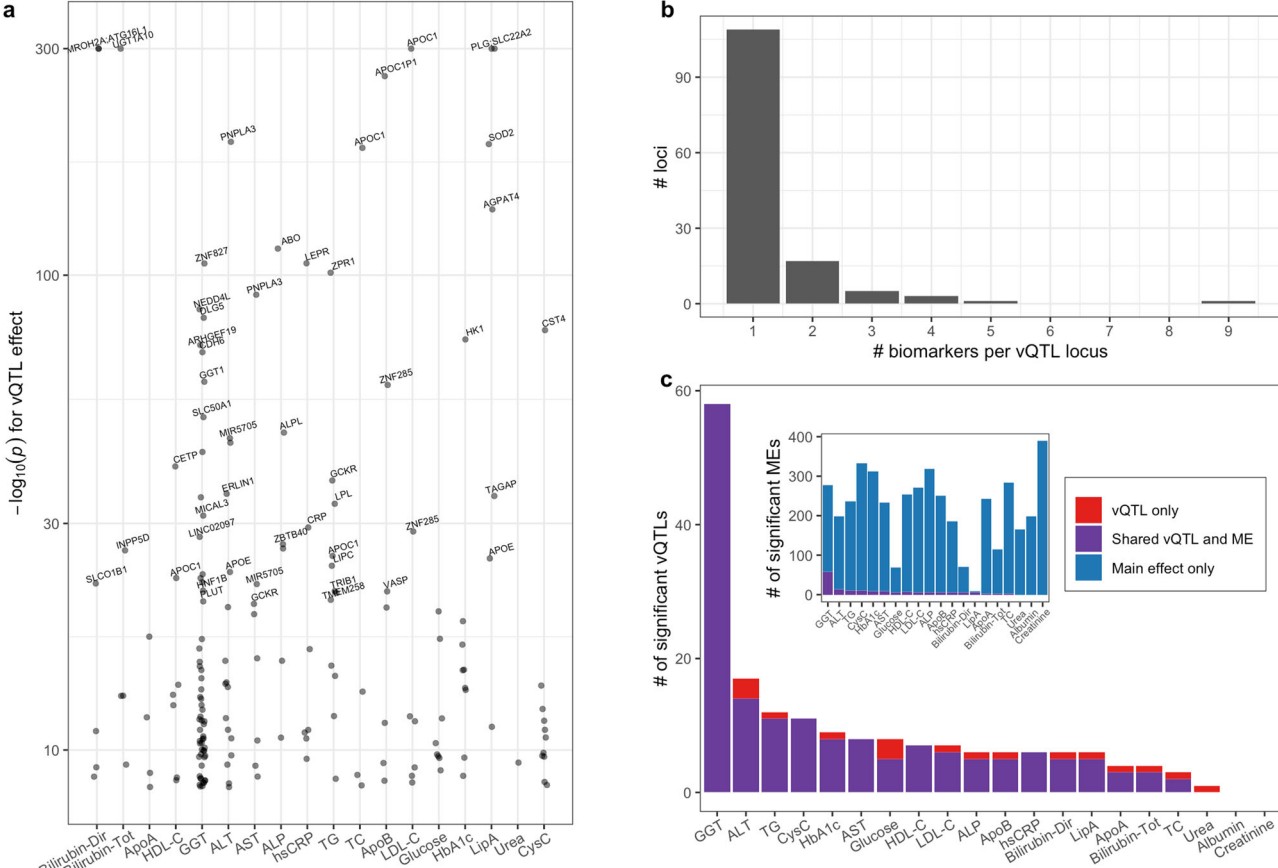

**Fig. 2 vQTLs identified across 20 cardiometabolic serum biomarkers. a** $-\log_{10}P_{vQTL}$ from Levene's test is shown for all significant index variants for each biomarker. Labels correspond to the closest gene (shown for variants with $P_{vQTL} < 10^{-20}$), highlighting some known GWAS loci. P values are truncated at $10^{-300}$ for visualization purposes. **b** Histogram displaying the number of biomarker associations for each vQTL locus. **c** The number of significant vQTL loci is shown for each biomarker (inset: analogous plot for main effects). Colors denote three categories: vQTL loci not shared with an ME locus (red), vQTL loci shared with an ME locus (purple), and ME loci not shared with a vQTL locus (blue).

associations. To induce a vQTL effect in the absence of a significant ME, genotypic effects on phenotype may have opposite signs in some subset of the population. In some instances, this may be explained by an underlying qualitative interaction (see the alcohol-*ADH1B* example below), while in other instances the relevant stratifying trait may remain unknown. Thus, rather than being spurious, at least some subset of these 17 vQTLs appears to truly lack overall MEs.

We next conducted a genetic correlation ($\rho_g$) analysis, using bivariate LD-score regression, to understand whether specific pairs of biomarkers were notably more or less similar in their genetic architecture when measured using MEs (standard) or vQTLs. We observed similar genetic correlations between biomarkers when using vQTL ($\rho_{g,vQTL}$) or main-effect ($\rho_{g,ME}$) summary statistics (Spearman correlation of 0.69 between $\rho_{g,vQTL}$ and $\rho_{g,ME}$ values across all non-identical biomarker pairs; Supp. Fig. S4). The mean $|r_{g,vQTL}|$ and mean $|r_{g,ME}|$ were also equal (0.16); therefore, while the associated significances of $r_{g,ME}$ tended to be substantially stronger, likely due to the known larger standard errors in variance estimation compared to mean estimation[16], the genetic correlation magnitudes tended to be similar. For three biomarker pairs, the $\rho_{g,vQTL}$ estimate was substantially different ($|\rho_{g,vQTL}-\rho_{g,ME}| > 0.2$ with nominal significance for both estimates. For example, one of these involved a greater $\rho_{g,vQTL}$ between HbA1c and ALT; these biomarkers may thus be more similar in their modifiable genetic effects (through GEIs, for example) than their fixed genetic effects.

We performed a replication analysis for each of the significant vQTLs for the 10 biomarkers available in the Women's Genome Health Study (WGHS; $N = 20{,}852$ women of European ancestry)[22]. Of 60 significant vQTL-biomarker pairs for which replication was possible, nominal replication ($p < 0.05$) was seen for 27 (45%) in spite of the much smaller sample size (full set of vQTL replication results in Supplementary Data 6). We found a strong correlation between discovery and replication significances (Spearman correlation of 0.51 between the p-values; Supp. Fig. S5a). The strongest vQTL associations in both the discovery and replication were with lipid-related biomarkers; for example, we find nominal replication for 5/7 LDL-C vQTLs, but only 1/9 HbA1c vQTLs.

**Exposome-wide interaction study reveals interactions underlying many vQTLs.** In stage two, we conducted exposome-wide interaction tests for the 182 vQTL-biomarker pairs. We used the PHESANT program to produce 2380 filtered and cleaned exposures, including physical, lifestyle, psychosocial, medical, and other types of traits (Supplementary Data 7, 8; see Methods). For each vQTL-biomarker pair, we tested for GEI effects involving the index variant and biomarker with each of 2380 exposure variables as potential effect modifiers, first stratified by ancestry and then meta-analyzed. Using a conservative multiple testing adjusted significance threshold ($p < 0.05/182/1156.2$ effective exposures $= 2.35 \times 10^{-7}$) we identified 847 significant interaction effects at 49 of the 184 vQTL-biomarker pairs, altogether representing 30 loci (Fig. 3, Supplementary Data 9).

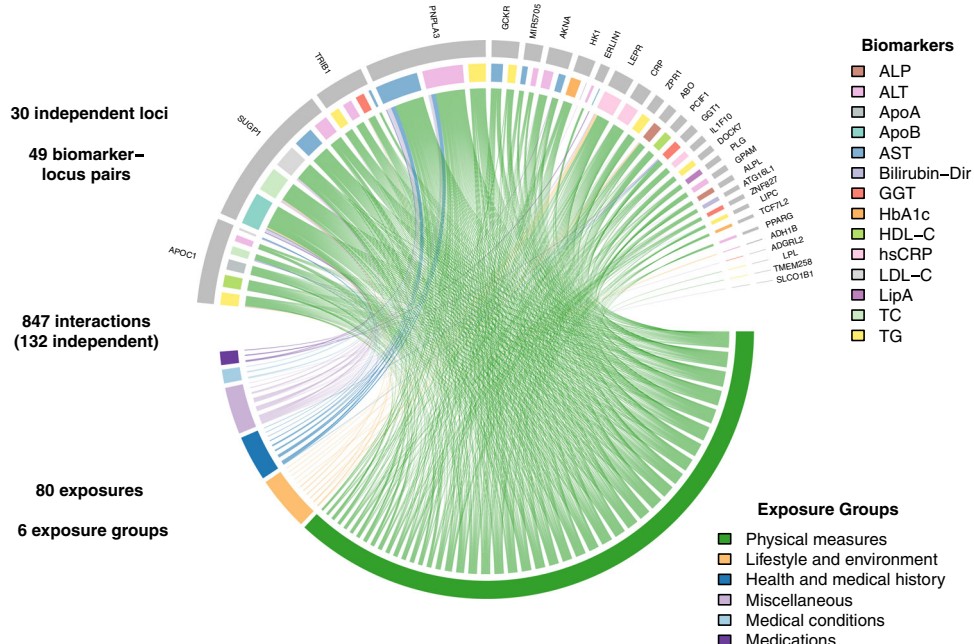

**Fig. 3 Chord diagram displays GEI links between vQTL-biomarker pairs (top of circle) and exposures (bottom of circle).** Lines correspond to interactions that are Bonferroni significant ($p < 2.38 \times 10^{-7}$ for the associated variant, biomarker, and exposure based on two-sided $p$-values from linear regression. vQTL-biomarker pairs are colored according to biomarker and labeled with the nearest gene. Exposures are colored according to exposure categories.

To understand the power gained for GEI discovery with a vQTL screening approach versus the more conventional approach prioritizing ME loci, we conducted a full EWIS using the same analytical pipeline on the larger set of stage one ME signals and compared GEI enrichment. Replicating findings from previous studies[15,16] with a much large set of exposures, we found substantial enrichment for GEIs when starting with signals prioritized from vQTLs versus MEs. Specifically, we found that 0.20% (847/437,920) of vQTL-biomarker-exposure triplets had a significant GEI versus 0.02% (1590/10,502,940) of ME-biomarker-exposure triplets (Chi-square test $p < 10^{-300}$). This 10-fold discovery enrichment remained after collapsing exposures and comparing variant-biomarker pairs; 26.6% (49/184) of vQTL-biomarker pairs had at least one significant GEI, versus just 3.3% (145/4413) of ME-biomarker pairs (Chi-square test $p = 1.8 \times 10^{-52}$). Of note, the EWIS using ME loci used the same Bonferroni-adjusted significance threshold as the EWIS on vQTL signals; if using MEs in practice, the more stringent Bonferroni adjustment would further reduce the number of GEIs identified.

The 847 GEIs were unevenly distributed across biomarkers, with greater than one hundred for ALT, TG, and AST and none for albumin, creatinine, cystatin C, random glucose, total bilirubin, or urea. Many of the participating exposures were from highly correlated categories such as anthropometric measures. To address this, we iteratively conditioned GEI tests on the most-significant exposure interaction at each locus to estimate the number of independent GEIs. We found 66 independent GEIs when retaining the same strict Bonferroni threshold ($p < 2.35 \times 10^{-7}$) for "secondary signals", which increased to 132 GEIs when using a nominal threshold ($p < 0.05$) for these secondary signals. However, this dense correlation structure also provides a unique opportunity to isolate specific relevant exposures with greater precision.

Returning to the WGHS dataset, we chose to replicate specifically those interactions involving BMI as an exposure, since (1) many significant GEIs implicated BMI or another anthropometric trait (89%), (2) BMI is an objective and standard

quantitative measurement that is easily compared across studies, and (3) BMI has strong biological links to the majority of biomarkers assessed in this study. Of 11 interactions involving BMI and one of the 10 available biomarkers in WGHS, nominal replication ($p < 0.05$) was seen for five interactions (55%; Chi-square $p = 4.71 \times 10^{-14}$ Supplementary Data 10). These included 3/5 (60%) BMI interactions for TG, 3/3 (100%) for hsCRP, and 0/1 for each of TC, HDL-C, and HbA1c. The replication of these interactions demonstrates their robustness across populations despite the substantially lower sample size in WGHS. As in the WGHS vQTL replication in stage one, we found that the strongest GEI signals by p-value from the discovery UKB cohort tended to be those replicated in WGHS.

**vQTLs and interactions are robust in sensitivity analyses**. To increase confidence in the catalog of vQTLs and interactions identified, we undertook a series of sensitivity analyses. First, we re-tested a subset of 179 vQTLs (those that were significant in the European-ancestry subset) using inverse-normal transformed (INT) biomarkers again in the European subset to confirm that the vQTLs were not artifacts of skewed biomarker distributions. While a substantial number of these relationships were attenuated, 47% remained significant at the Bonferroni level (94% at a nominal threshold of $p < 0.05$; Supp. Fig. S6).

Previous work has shown that naive two-stage, vQTL screening-based interaction testing procedures can have inflated type I error whenever the exposure tested in stage two is associated with the outcome, due to a correlation between test statistics for stage one and stage two[23]. In the context of a single exposure, one solution is straightforward: residualize the phenotype on the exposure prior to vQTL testing. However, this approach is not optimal when undertaking an unbiased EWIS with over 2000 exposures: it is impractical to stably fit regression models with this dimensionality in the smaller non-European ancestry groups in this study ($N < 6100$). Thus, we performed a sensitivity analysis based on the results of the primary analysis,

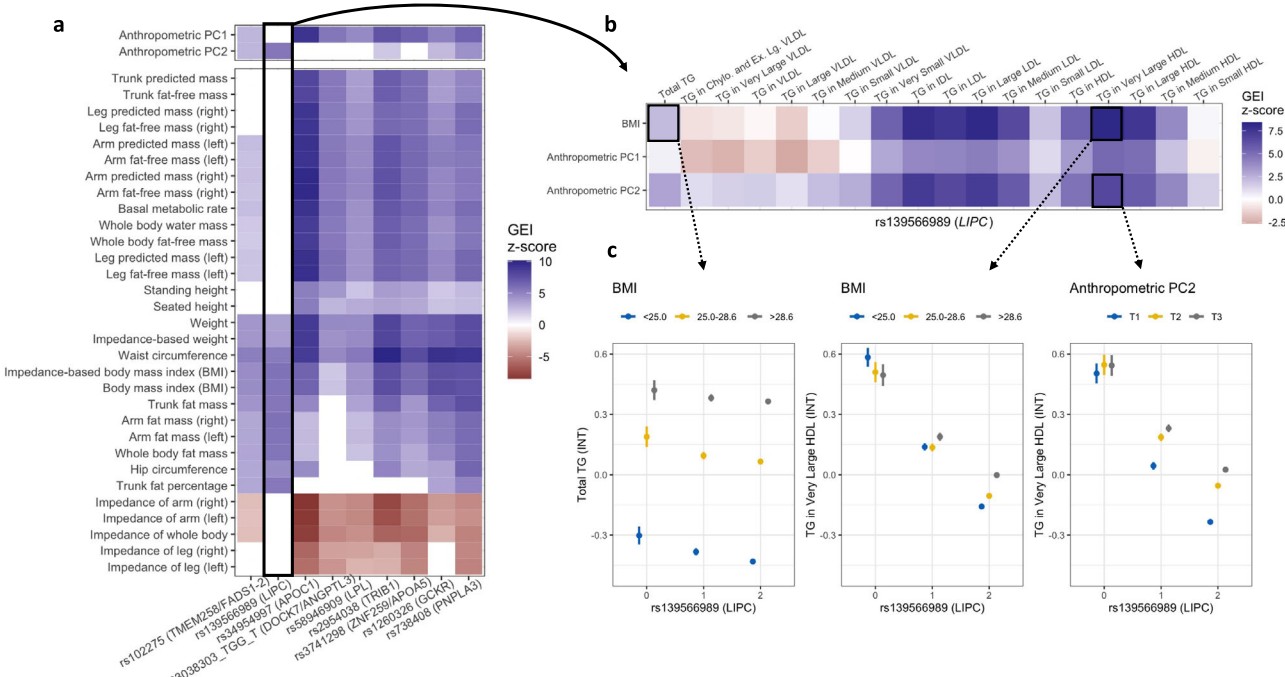

**Fig. 4 Exploration of anthropometric interactions influencing triglycerides. a** Heatmap shows interaction z-scores from standard linear regression models between nine genetic variants (*x* axis) and 33 anthropometric exposures (*y* axis). Colored panels pass a nominal significance threshold (*p* < 0.05). Variants are annotated with the closest gene, as well as a second likely causal gene based on manual annotation where appropriate. **b** Heatmap shows interaction *z* scores for a single variant (rs139566989 in *LIPC*), with varying TG lipid subfraction outcomes from nuclear magnetic resonance (*x* axis) and three representative anthropometric exposures (*y* axis). **c** Three stratified plots showing means and 95% confidence intervals for inverse-normal transformed total TG or TG in very large HDL after stratification by rs139566989 and tertiles of the relevant exposure (labeled at the top of each plot; *n* = 90,644 total participants).

residualizing each pre-processed biomarker on the much smaller set of 88 significant GEI exposures (separately in each ancestry) and re-performing vQTL tests plus meta-analysis at the loci of interest. In these exposure-adjusted analyses, 140 out of the original 184 vQTL-biomarker pairs (76%) remained significant (Supp. Fig. S7a), with only one completely diminished signal (*p* > 0.05). This result means that most of these vQTLs would have passed on to stage two for interaction testing even after explicit removal of this potential bias in the two-stage testing procedure. As additional support for the robustness of our interaction results in false positives, we did not observe substantial systematic inflation of interaction *p* values across the entire set of EWIS tests (Supp. Fig. S7b).

**Investigation of anthropometric exposures for triglyceride GEIs highlights distinct biology**. Many significant GEIs involved correlated anthropometric exposures, such as BMI, waist and hip circumference, and bioelectrical impedance metrics (which measure body fat and fat-free mass). Focusing on TG specifically as a biomarker with known complex relationships with obesity and cardiometabolic disease risk, we extracted GEI *z* scores for the nine variants participating in significant interactions with an anthropometric trait (all of which are annotated to well-known TG-related genes). The nine variants showed distinct patterns of variation across exposures: two interacted more strongly with body fat measures (*FADS1–2* and *LIPC*), three interacted more strongly with body mass measures (*APOC1*, *ANGPTL3*, and *LPL*), and four were balanced across anthropometric exposures (*TRIB1*, *APOA5*, *GCKR*, and *PNPLA3*) (Fig. 4a). Given such highly correlated measurements, we also conducted a principal components analysis on the 33 relevant anthropometric exposures in the

European subset, finding that the top principal components appeared to represent body mass (PC1) and body fat (PC2; Supp. Fig. S8). These PCs reproduced the differential interactions with body mass and body fat (Fig. 4a).

The intronic indel rs139566989 in the hepatic lipase (*LIPC*) gene interacted almost exclusively with body fat-related measures. Hepatic lipase is a glycoprotein in the triacylglycerol lipase family that is important for the metabolism of lipoproteins including intermediate-density lipoprotein (IDL) and HDL[24]. The insertion (AF = 79% in the European subset) is associated with increased expression of *LIPC* (Genotype-Tissue Expression Portal browser) and decreased serum TG ($p = 3.86 \times 10^{-59}$ in our primary ME analysis). Its GEI effect with BMI represents a small but significant increase in the positive association between BMI and serum TG ($p$ interaction = $6.05 \times 10^{-9}$). To further pinpoint relevant biological mechanisms, we further tested GEIs with BMI and the two anthropometric PCs for their effects on TG-containing lipid subfractions measured by nuclear magnetic resonance in the UKB (N~90,000; Fig. 4b). We found the strongest interactions (with both BMI and anthropometric PC2) for TG in IDL, large LDL, and large HDL (Fig. 4b). For example, the insertion increased the association between adiposity and TG in very large HDL: the bottom and top BMI tertiles had little TG difference in non-carriers, but a mean difference of almost 0.3 standard deviations in insertion homozygotes (Fig. 4c). Interactions between these three primary anthropometric exposures and all nine TG-related variants impacting TG subfractions can be found in Supplementary Data 11. Though it does not impact the relevance of these findings, we note that rs139566989 may not be causal; a fine-mapping analysis from a trans-ancestry GWAS for lipids identified a nearby variant, rs1077834 ($r^2 = 0.90$ with rs139566989 in the 1000 Genomes Project) as the most likely causal variant for TG in this locus[25].

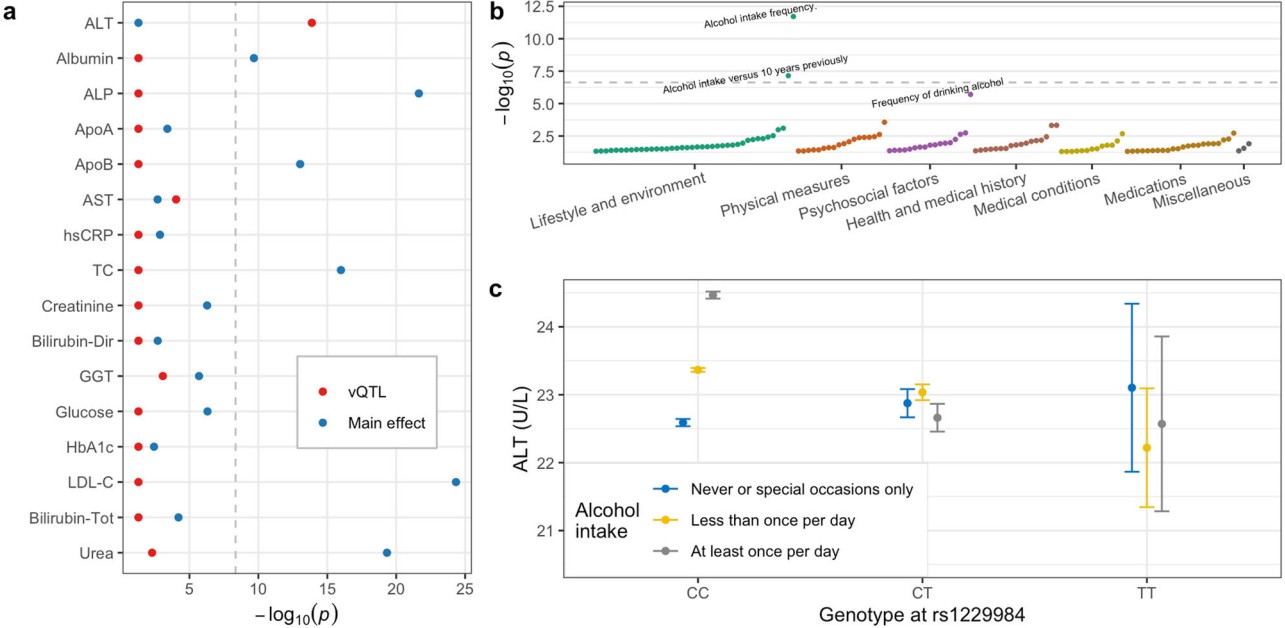

**Fig. 5 vQTL and GEI relationships for ADH1B, alcohol, and ALT. a** vQTL (red) and ME (blue) significance for rs1229984 is shown for each biomarker, based on Levene's test and linear regression, respectively. The dashed line represents the study-wide Bonferroni vQTL significance threshold. Biomarkers with neither vQTL nor ME having $p < 0.01$ are not shown. **b** EWIS results for rs1229984 impacting ALT are shown, with GEI significance plotted (y axis) for each exposure having $p < 0.05$ (x axis). **c** Means and standard errors for ALT ($n = 341,815$ [CC], 17,539 [CT], 318 [TT] participants) are plotted as a function of genotype at rs1229984 (x axis) and self-reported alcohol intake (colors).

A group of three variants (annotated to *APOC1*, *ANGPTL3*, and *LPL*) had a strong interaction with body mass PC1 on TG levels (all $p \leq 2.4 \times 10^{-6}$) but none with body fat PC2. These three genes are involved in the production or regulation of lipoprotein lipase, which cleaves TG from circulating lipoproteins[24]. These body mass-specific interactions may reflect (1) biological processes in skeletal muscle rather than adipose, or (2) body mass-associated behavioral characteristics such as total caloric intake, which is likely better measured by body mass than from self-reported questionnaires.

**Alcohol intake interacts with a common ADH1B polymorphism to influence liver stress.** One of our lead vQTLs with a clear underlying GEI was the combined effect of SNP rs1229984 and alcohol consumption on ALT, a biomarker of liver stress. SNP rs1229984 is a missense variant in the *ADH1B* gene that affects alcohol processing in the liver and is known to influence alcohol consumption[26]. This variant had a strong vQTL effect ($p = 1.31 \times 10^{-14}$) but did not have a significant main effect ($p = 0.031$) in the meta-analysis for ALT (Fig. 5a), highlighting the value in the vQTL screen for stage 1. In the following exposome-wide scan, the exposure most strongly interacting with rs1229984 to influence ALT was in fact self-reported "alcohol intake frequency" ($p = 1.9 \times 10^{-12}$ Fig. 5b). After stratifying self-reported alcohol intake into three bins within the European-ancestry group, we observed a substantial positive association between alcohol and ALT in homozygous major allele carriers, versus no such increase for minor allele carriers (Fig. 5c). This interaction also demonstrates how a strong vQTL effect can arise with a minimal overall ME: rs1229984 is associated positively with ALT in the never or rare alcohol consumers, but negatively in frequent consumers (Fig. 5c), thus "canceling each other out" in the marginal ME test. The rs1229984-ALT vQTL was robust to pre-adjustment of the pre-processed ALT distribution for alcohol intake ($p = 6.55 \times 10^{-13}$ in the European-ancestry subset after adjustment), confirming that this original vQTL signal was not

simply an artifact of the alcohol-ALT relationship. This interaction suggests that alcohol consumption may be of greater concern for potential liver damage in individuals homozygous for the major allele at rs1229984. We note that mean alcohol intake was much lower in T allele carriers, producing a decrease in intake variability that may contribute to the lack of alcohol-ALT relationship in these individuals.

## Discussion

We conducted a two-stage analysis to first identify vQTLs impacting cardiometabolic traits then conduct a systematic search for underlying GEIs. Using this strategy, we identified 136 loci associated with the variance of one or more of 20 cardiometabolic serum biomarkers, subsequently uncovering 847 GEIs impacting those biomarkers at 30 independent loci.

Results from our study largely align with previous results from Wang and colleagues, which utilized analogous methods in the same population (European-only subset of UKB)[15]. Examining anthropometric and lung function traits, they found a similar average number of vQTL relationships per effective phenotype (15 versus 16.6 in our study), indicating a similar level of polygenicity of vQTL effects. They found a somewhat smaller enrichment of vQTLs in ME loci (1.7% of ME loci contained vQTLs versus 3.7% in our study), suggesting that the genetic control of cardiometabolic biomarkers may be particularly susceptible to modulation by genetic background or environmental exposures compared to traits like height or forced vital capacity.

We observed that vQTLs were predominantly found in ME loci and that their strengths of association were generally correlated with the strengths of corresponding main effects. This observation has three important implications. First, it impacts the value of vQTLs as a screening tool. Screening that simply uses genetic main effects is a common and viable strategy[8], but we note that (1) the presence of a vQTL further increases the likelihood of identifying an underlying interaction[16], (2) in our analysis, the use of vQTLs reduced the genetic search space by more than an

order of magnitude compared to genetic MEs, and (3) the alcohol-*ADH1B* interaction is an example of a strong interaction found via vQTLs that would not have been explored using only ME-based screening. Second, the correlation between vQTL and ME strengths could theoretically suggest that the vQTLs are statistical artifacts unrelated to underlying interactions. Non-normally distributed variables have a mean-variance relationship and thus some vQTL tests are susceptible to false positives in the presence of genetic MEs and can be sensitive to trait transformations[14,27]. However, previous studies have demonstrated reasonable robustness of Levene's median-based test to non-normality[15] and our sensitivity analyses demonstrate that inverse-normal transformed phenotypes, for which the correlation between mean and variance effects is reduced, produce largely similar vQTL results. Third, it highlights the weight of prior probability in agnostic analyses: loci that have already demonstrated a role in biology (by being implicated in MEs) are more likely to be relevant in GEI searches as well.

Our two-stage approach required $4.4 \times 10^5$ GEI tests, compared to the $4.8 \times 10^{11}$ tests that would have been necessary to exhaustively explore all genetic variants and biomarkers. Beyond the clear practical and computational benefits of this reduction in the genetic search space, the statistical value in this two-stage analysis is dependent on both the power of Levene's test and the extent to which it decreases the multiple testing penalty in the second stage (in this study, $p < 2.35 \times 10^{-7}$ compared to $p < 3.90 \times 10^{-12}$ without prioritization). While it is well-established that vQTL loci are more enriched than ME loci for underlying GEIs[15,16], the power of Levene's test may be affected by the strength of the underlying exposure-outcome relationship and may be compromised in cases where a complex set of underlying interactions exist without consistent directions of effect[11]. Given this fact, it is not surprising that many of our significant GEIs involved straightforward and high-impact exposures such as body size (e.g., BMI) and alcohol intake as compared to more-complex traits like socioeconomic status indicators or dietary behaviors.

The PHESANT tool, which processes thousands of phenotypes in an automated way, was developed to facilitate phenome-wide association studies[6], but here we leveraged it to create a library of exposures to be used in GEI tests. Many of its motivations apply equally in this context—non-normally distributed continuous variables and highly imbalanced binary variables can create instability and bias in GEI tests (as exposures) just as in standard genetic main effect tests (as outcomes). PHESANT enabled the high-throughput pre-processing of 2380 variables for our exposome-wide analysis, which would be impractical to prepare one-by-one. We note that the semi-automated nature of this process means that there may be more appropriate quality control and coding strategies for specific exposures of interest in follow-up studies.

Our approach identified 847 significant GEIs, many of which involved a set of highly correlated exposures (discussed further below). The large sample size of UKB provides the statistical power needed to identify interactions and has supported genome-wide GEI discovery in investigations of anthropometric and cardiometabolic phenotypes[5,28–30]. However, the scale of this population also means that spurious GEIs induced by even small statistical artifacts or biological confounders may reach statistical significance, as demonstrated by Tyrell and colleagues[31]. Thus, the catalog of GEIs described here should be treated as hypothesis-generating rather than confirmatory.

Many of the significant GEIs involved a set of highly correlated anthropometric exposures (e.g., BMI and waist circumference). These are not typically used as GEI exposures because they are not behavioral or environmental, but they can nonetheless play a

role in modifying genetic associations with cardiometabolic biomarkers and outcomes[32]. The dense correlation of these exposures reduces the number of effective discoveries, but this more-comprehensive coverage of the exposure space allowed us to tease apart underlying mechanisms. Focusing on TG as an outcome, we found that nine relevant variants tended to fall into three groups, interacting with anthropometric measures of either body mass, body fat, or both. We further validated this observation using representative principal component summary variables. For one variant (rs139566989 in *LIPC*), body fat interactions affected TG in specific lipoprotein subfractions such as IDL and very large HDL, consistent with the well-established role of hepatic lipase in lipoprotein remodeling (especially the IDL-to-LDL and large-to-small HDL conversions)[24]. Lipoprotein-related GEIs involving *LIPC* variants have been previously reported for relevant exposures such as dietary fat composition in weight loss trials[33,34] and observational cohorts[35]. In general, our fine-grained exposure analysis provides insight into lipoprotein biology and highlights the importance of comprehensive and precise phenotyping.

A primary strength of this investigation is the comprehensiveness of cardiometabolic biomarkers and exposures examined. However, the pipeline relies on many self-reported exposure measurements; follow-up of specific interactions should employ more objective and precise methodologies (e.g., accelerometer-based physical activity measurements) as well as more comprehensive sensitivity analyses to refine interaction estimates. Additionally, we were only able to assess replication of vQTLs and GEIs for half of the biomarkers of interest. While the replication analysis provides confidence in the general robustness of our approach across populations, many signals from UKB, such as those for liver enzymes, bilirubin, and random glucose, were not able to be tested in WGHS. The WGHS also contains only women, which may particularly impact replication for sex-differentiated biomarkers such as HDL-C. Finally, while multiple ancestries were analyzed here as available, the meta-analysis results primarily reflect the effects on European individuals due to the major sample size discrepancy. Future work should incorporate more ancestry-balanced data sets, explore the heterogeneity of interaction effects across ancestries, and pursue follow-up of some of the many ancestry-specific vQTLs and interactions described here.

The vQTLs and GEIs described here have been made publicly available in the AMP Common Metabolic Disease Knowledge Portal (https://hugeamp.org/research.html?pageid=UKB-vQTL-GxE). These catalogs can generate hypotheses for future research and enable lookups to better characterize findings from GWAS and other genetic studies. Our findings extend previous proofs-of-concept of the vQTL strategy to the realm of cardiometabolic biomarkers, establish a set of loci to be prioritized for further GEI analysis, and highlight specific GEIs that can inform cardiometabolic precision medicine strategies.

## Methods

**UK Biobank genetic data**. This work was conducted under a Not Human Subjects Research determination for UKB data analysis (NHSR-4298 at the Broad Institute of MIT and Harvard). UKB is a large prospective cohort with both deep phenotyping and molecular data, including genome-wide genotyping, on over 500,000 individuals ages 40–69 living throughout the UK between 2006–2010[36]. Genotyping, imputation, and initial quality control on the genetic dataset have been described previously[37]. We removed individuals flagged for failing UKBiLEVE genotype quality control, heterozygosity or missingness outliers, individuals with putative sex chromosome aneuploidy, individuals with self-reported vs. genetically inferred sex mismatches, and individuals that had withdrawn consent by the time of analysis. Additionally, we subset to a group of unrelated samples by including only those that were used for genetic principal components analysis (PCA) during central genetic data pre-processing[37]. Furthermore, only genetic variants with minor allele frequency (MAF) > 0.005 in the full sample were included in the present analysis (in addition to subsequent ancestry-specific MAF filters).

Ultimately, 9,487,345 variants were used for analysis in the European-ancestry subset, with slightly different numbers in the three other ancestry groups based on the MAF filter. For the vQTL and GWAS analyses, imputed genotypes within 0.1 of an integer value (0, 1, or 2) were converted to hard-calls using PLINK2[38] and all other values were set to missing. Work was conducted on genetic data release version 3, with imputation to both Haplotype Reference Consortium and 1000 Genomes Project (1KGP), under UK Biobank application 27892.

UKB samples were grouped into four primary ancestry groups: African (AFR), East Asian (EAS), European (EUR), and South Asian (SAS). We defined ancestry groups using a Gaussian mixture model on the principal components of 1000 Genomes project (1KGP) phase 3 data. The Gaussian mixture model is a probabilistic model for representing normally distributed subpopulations within an overall population, by assigning data points to the multivariate normal components that maximize the component posterior probability given the data using the expectation-maximization algorithm[39]. Each component mean was defined by selecting data points randomly without replacement at initialization. We built the Gaussian mixture models using the Scikit-learn Python package and tested 10-fold cross-validation on 40 models with different initialization states. We used the adjusted rand index score[40] to evaluate the similarity between predicted and true clustering using 1KGP principal components and their 1KGP ancestry labels. This approach was used to refine the subpopulation clustering of 1KGP principal components. We then projected UKB principal components onto the 1KGP principal component space and used the Gaussian mixture model with the highest adjusted Rand index score to cluster individuals into ancestry subgroups.

**Serum biomarkers**. We focused on 20 serum biomarkers related to cardiovascular disease and metabolism, including but not limited to lipids, liver enzymes, glycemic parameters, and kidney function markers (see Supplementary Data 2). Blood samples were collected at the baseline visit for the majority of participants, and specific biomarkers were measured using colorimetric, enzymatic, and immunoassays (details available at: https://biobank.ctsu.ox.ac.uk/crystal/crystal/docs/serum_biochemistry.pdf).

Phenotype data processing and other downstream analyses were conducted using R version 3.6.0[41] unless described otherwise. Biomarker values were pre-processed separately within each ancestry group, and followed a modified version of the procedure described by Sinnott-Armstrong and colleagues; see original manuscript for detailed description and online code[42]. We additionally removed individuals with diabetes, coronary heart disease, cirrhosis, end-stage renal disease, or cancer diagnosis within one year prior to their assessment center visit, or who were pregnant within one year of the assessment center visit. Briefly, log-transformed biomarkers were adjusted for sex, age, self-identified ethnicity, fasting time, dilution factor, assessment center, genotyping batch, genetic principal components, time of sampling, month and day of assessment, and a series of interactions between covariates. Log-transformation was chosen because it most closely mirrored the GWAS by Sinnott-Armstrong et al. and because initial empirical results showed substantial inflation of vQTL p-values for highly-skewed raw biomarker values. Of note, cholesterol, LDL-C, and Apolipoprotein B were also adjusted for statin use using methods described previously[42]. In individuals with self-reported use of a statin medication, each of these biomarkers was divided by an adjustment factor (0.749, 0.684, and 0.719, respectively) that had been empirically estimated by Sinnott-Armstrong and colleagues in the same population. Following residualization, outliers (residuals >5 standard deviations from the mean) were set to missing. Finally, residuals were z-score normalized to mean zero and variance one.

Due to the substantial correlation between the 20 cardiometabolic biomarkers, a smaller number of "effective" biomarkers was calculated to inform multiple hypothesis testing corrections. Using a previously-described approach[43], pre-processed biomarkers were collected into a single dataset across ancestries and principal components analysis was performed (*prcomp* function with standardized variables). The number of effective biomarkers was then calculated from the principal component variances $\lambda$ (equal to the eigenvalues of the biomarker covariance matrix) as $N_{BM,eff} = \frac{(\sum_{k=1}^{p} \lambda_k)^2}{\sum_{k=1}^{p} \lambda_k^2}$.

**Variance-QTL analysis**. vQTL analysis was performed using the vQTL module from the OSCA suite[44]. Differential variance tests used the median-based Levene's test, which is equivalent to a one-way analysis of variance (ANOVA) for absolute deviations from the median biomarker value. The test statistic, which is described in detail elsewhere[15], is:

$$\frac{(n-k)}{(k-1)} \frac{\sum_{i=1}^{k} n_i (z_{i.} - z_{..})^2}{\sum_{i=1}^{k} \sum_{j=1}^{n_i} (z_{ij} - z_{i.})^2} \qquad (1)$$

where $i$ is the group indicator (one of three genotypes), $n_i$ is the sample size for the $i$th genotype, $z_{ij}$ is the absolute difference between the phenotype value in sample $j$ from genotype $i$ and the median value in genotype $i$, $z_{i.}$ is the average $z$ value in genotype $i$, and $z_{..}$ is the average $z$ value across all samples. Under the null hypothesis of equal variances, this statistic follows an $F$ distribution with $k-1$ and $n-k$ degrees of freedom.

Genome-wide vQTL analysis was performed separately for each of the 20 biomarkers (Supplementary Data 2) in each of the four ancestry groups. Multi-

ancestry meta-analysis was then performed using METAL (2011-03-25 version)[45]. Stouffer's method (*SAMPLESIZE* mode) was used to perform fixed-effects meta-analysis based on p-values, sample sizes, and effect directions. While Levene's test does not natively produce effect sizes or directions, they are derived by the OSCA tool based on a method described by Zhu and colleagues[46] and previously implemented for vQTLs[15]. An effect direction is determined by regression of absolute deviations from the median on additively-coded genotypes, then this sign is combined with the Levene's test p-value to derive an effect size and standard error through back-transformation: $b = \frac{z}{\sqrt{2p(1-p)(n+z^2)}}$ and $SE = \frac{1}{\sqrt{2p(1-p)(n+z^2)}}$ where $b$ is the vQTL effect size, $z$ is a z-statistic derived from the $p$ value and effect direction, $p$ is the minor allele frequency of the variant, and $n$ is the sample size.

For each genome-wide vQTL scan (20 biomarkers × 4 ancestries), we performed an analogous standard GWAS testing for genetic effects on the mean of the phenotype using the *–glm* command in PLINK2[38]. These GWAS used the same adjusted phenotypes with no additional covariates, followed by meta-analysis as described above but using effect size and standard error-based meta-analysis (*STDERR* mode).

Genome-wide summary statistics were pruned by removing variants within 500 kb of each index variant. This distance-based procedure was chosen to be reasonably conservative and avoid the need for separate LD matrices per ancestry. To identify overlapping variants across biomarkers, ancestries, and analysis types (vQTL vs. ME), relevant pruned summary statistics were combined and a second clumping procedure (similarly defined based on 1MB windows) was performed. For example, to identify overlapping vQTL loci across ancestries for ALT, all ancestry-specific, pruned summary statistic matrices for ALT (four in total) were stacked and then subject to a similar iterative clumping procedure as was used for the initial pruning.

Genetic correlation analysis was performed using bivariate LD-score regression[47,48] for both vQTL and ME results from the European-ancestry subset. For each pair of biomarkers, summary statistics were retrieved and analyzed along with a European-ancestry linkage disequilibrium reference dataset from the 1000 Genomes Project (https://alkesgroup.broadinstitute.org/LDSCORE/).

Differential enrichment for epigenomic features between vQTL and ME loci was assessed using the Locus Overlap Analysis (LOLA) strategy and associated R package[21]. LOLA calculates enrichment of a set of target regions of the genome for enrichment in a set of epigenomic annotations compared to a background "universe" set of loci. The basic "LOLA core" set of epigenomic annotation collections was used. The background locus set was chosen to be the union of the vQTL and ME loci such that the resulting significance tests directly compared enrichment between the two groups for any given epigenomic annotation. Significance was assessed after the application of a Bonjamini–Hochberg false discovery rate correction.

**Preparation of exposure variables**. In order to conduct the GEI analysis in an exposome-wide manner, we needed to collect and clean a large set of relevant exposure variables. For this purpose, we used a program for automated phenotype pre-processing initially developed for phenome-wide association studies (PHEnome Scan ANalysis Tool, or PHESANT[6]) and later expanded (https://github.com/astheeggeggs/PHESANT). This PHESANT procedure relies on files describing variable input types (continuous, integer, categorical single [one choice], and categorical multiple [multiple choice]) and data coding schemes. Based on this input, it then conducts automated pre-processing of the phenotypic dataset, including removal of variables with excessive missingness or insufficient heterogeneity, conversion of categorical variables to ordinal and binary variables, and inverse-rank normal transformation of continuous variables.

As originally applied, the PHESANT pipeline processed a comprehensive set of phenotypes appropriate for phenome-wide association studies. We modified the set of included variables to better reflect the space of traits and exposures that may participate in GEIs. For example, sex does not make sense to use as an outcome, but may be an important characteristic that modifies genetic associations with blood biomarker levels. Specifically, in comparison to the PHESANT pipeline referenced above, we additionally included sex, age, and assessment center, and 24-hour recall-based dietary assessments and excluded blood-based assays, hospital records, and cancer and death registers. After running PHESANT on the full UK Biobank dataset, the output dataset contained 2380 variables for analysis. We note that this includes two variables both tracking participant age based on the particulars of the PHESANT functionality. This corresponded to 1,156.1 effective exposures tested based on the same procedure used above for determining the number of effective biomarkers. These exposures included physical (e.g., BMI), lifestyle (e.g., dietary behaviors), psychosocial (e.g., neuroticism score), medical (e.g., statin use), and other types of traits. Based on the data dictionary available from UKB and additional manual curation, phenotypes were placed into one of eight exposure categories. Exposure group summary counts and variable examples are available in Supplementary Data 7, while a full list of exposure variables is available in Supplementary Data 8.

**Genome-wide interaction studies**. For each combination of vQTL-biomarker pair (from Stage I) and exposure (from PHESANT), a GEI test was performed using the following simple model:

$$BM \sim g + e + g * e \qquad (2)$$

where $BM$ represents the pre-processed biomarker, $g$ represents the imputed genotype dosage at the vQTL index variant, and $e$ represents the relevant exposure. For the primary analysis, no additional covariates were used given the prior residualization on basic covariates (age, sex, and genetic principal components). Interaction analysis was performed with GEM v1.3[49] using robust standard errors. As with the vQTL and GWAS analyses, these interaction tests were performed separately in each ancestry group, followed by the inverse variance-weighted, fixed-effect meta-analysis[45]. Principal components analysis using prcomp in R was conducted on a set of anthropometric exposures in individuals of European ancestry in UK Biobank to include as additional body size-related exposures in GEI testing.

Because of the substantial correlation between many tested exposures, we conducted a follow-up analysis to determine the number of independent GEIs. For each vQTL-biomarker pair participating in a significant GEI from the primary EWIS (and thus significant at the strict Bonferroni threshold), we re-tested for GEI with each exposure. Then, we iteratively re-tested while additionally conditioning on the GEI effect for the most-significant exposure. We report the number of total independent GEIs, counting secondary signals using a nominal significance threshold ($p < 0.05$) and using our stricter EWIS Bonferroni threshold.

To compare enrichment of vQTL versus ME loci for underlying GEIs, we conducted a parallel EWIS using the larger set of ME loci. Other than the different set of input variants (ME index variants rather than vQTL index variants), this analysis was identical to the primary EWIS (including the use of the same Bonferroni threshold).

**Anthropometric exposure and lipid subfraction analysis**. Anthropometric exposures were defined as those with a "Level 3 Category" of "Anthropometry" in the UKB data dictionary. Initially, GEI summary statistics from the EWIS were collected for this set of exposures with TG as the outcome and including nine genetic variants passing the EWIS significance threshold for at least one of these 31 exposures. These nine variants were manually annotated to likely causal genes based on known lipid biology. To create a set of anthropometric summary variables for GEI analysis, PCA was performed on this set of exposures in the European ancestry subset. The first two principal components were determined to represent body mass and body fat based on inspecting trait loadings. These two components were extracted and used for GEI testing.

Triglyceride concentrations in various lipoprotein subfractions, based on nuclear magnetic resonance (NMR) spectroscopy metabolomics data, were retrieved for ~90,000 European ancestry individuals (differing numbers per metabolite based on missingness). Quality control was performed using measures provided by the UKBB, removing values lower than the limit of detection and samples with likely contamination or degradation issues. Metabolite measurements were inverse-normal transformed prior to modeling. GEI testing was performed for each anthropometric exposure-TG subfraction pair (as exposure and outcome, respectively), with additional adjustment for age, sex, 10 genetic PCs, NMR batch, and spectrometer.

**WGHS replication analysis**. The WGHS cohort was used to replicate vQTLs and BMI GEIs. WGHS is a prospective US-based cohort of healthy adult females 45 years or older[22]. The biomarkers available in WGHS included ApoA, ApoB, hsCRP, TC, creatinine, HbA1c, HDL-C, LDL-C, LipA, and TG, and were pre-processed and analyzed similarly to biomarkers in UKB (limiting to European ancestry individuals, log transforming, removing individuals with diabetes or on blood pressure medication, adjusting for covariates, and testing vQTL hard-call genotype associations using Levene's test in R; $N = 20,852$). The same pre-processed biomarkers were used for replicating the BMI GEIs, adjusting for age and genetic PCs. Effect directions were not considered when assessing vQTL replication in WGHS.

**Reporting summary**. Further information on research design is available in the Nature Research Reporting Summary linked to this article.

## Data availability

No new genetic or phenotypic data have been generated for this study. The UK Biobank data, including genetic and phenotypic data, are under restricted access for privacy reasons but can be obtained through application at https://www.ukbiobank.ac.uk/. UK Biobank will consider data applications from bona fide researchers for health-related research that is in the public interest. Genome-wide meta-analysis summary statistics for both vQTL and GEI results from this study are available for visualization and download at https://hugeamp.org/research.html?pageid=UKB-vQTL-GxE.

## Code availability

Code supporting the conclusion of this manuscript can be found at: https://github.com/kwesterman/vqtl-gxe.

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

## Acknowledgements

The authors would like to thank Jordi Merino of Massachusetts General Hospital and Chirag J. Patel of Harvard Medical School for helpful comments on the analysis. K.E.W. was supported by NIDDK Fellowship 2T32DK007028-46. D.K.J. is supported by NIDDK 2UM1DK105554 and NHGRI U24 HG011453. H.C. and A.K.M. are supported by NHLBI R01 HL145025. J.C.F. is supported by NHLBI K24 HL157960. M.S.U. is supported in part by NIDDK K23DK114551. J.B.C. is supported by NIDDK Pathway to Independence Award (K99DK127196). This research was conducted using the UK Biobank Resource under Application Number 27892. The WGHS is supported by the National Heart, Lung, and Blood Institute (HL043851 and HL080467) and the National Cancer Institute (CA047988 and UM1CA182913), with funding for genotyping provided by Amgen. The Genotype-Tissue Expression (GTEx) Project was supported by the Common Fund of the Office of the Director of the National Institutes of Health, and by NCI, NHGRI, NHLBI, NIDA, NIMH, and NINDS. The data used for the analyses described in this manuscript were obtained from the GTEx Portal on 10/18/2021. Although K.E.W. and A.K.M. are employees of Mass General Brigham, this work was not conducted in their capacity as Mass General Brigham employees.

## Author contributions

K.E.W., A.K.M., and J.B.C. designed the experiment. K.E.W. and J.B.C. conducted the primary analysis. T.D.M. conducted the UK Biobank metabolomic data analysis. H.C. and A.K.M. contributed to the design and interpretation of the two-stage analysis approach. F.G. and D.I.C. contributed to the replication analysis. J.C.F. provided UK Biobank data access. J.M. derived UKB ancestry labels. J.C.F. and M.S.U. provided clinical interpretation of findings. K.E.W., D.K.J., and J.B.C. contributed to the storage and visualization of these results in an online catalog. M.S.U., D.I.C., A.K.M., and J.B.C. supervised the analysis and provided critical revision of the manuscript. K.E.W. and J.B.C. wrote the manuscript. All authors reviewed and approved the final manuscript.

## Competing interests

K.E.W. has provided consulting services for FOXO BioScience. J.M. is now an employee at Thermo Fisher Scientific. All other authors declare no competing interests.
