## [Peer Review File · Nature Communications]

Variance-quantitative trait loci enable systematic discovery of gene-environment interactions for cardiometabolic serum biomarkersREVIEWER COMMENTS

Reviewer #1 (Remarks to the Author):

"Variance-quantitative trait loci enable systematic discovery of gene-environment interactions for cardiometabolic serum biomarkers" is a very interesting and well-written manuscript. Identifying gene environment interactions is difficult, and I am impressed by the number of interactions discovered by employing a two-stage, variance QTL based approach. The authors analyze their results to present several unique findings, such as teasing out triglyceride interactions with either body mass or body fat exposures, and a strong ADH1B-alcohol interaction despite no mean-based association at the ADH1B locus.

Overall, I am impressed by the sheer number of associations and several unique findings. While the technical analyses appear to be robust and valid, I would like to see the work strengthened by elaborating on their analyses and results. Particularly:

- (1) it is not clear how multi-ancestry aids the analysis,
- (2) I would like to see if the multi-cohort context could be further leveraged in vQTL analysis,
- (3) it is not clear what contrasts vQTLs from ME loci, and
- (4) how many independent interactions exist.

Addressing these points, which are described in greater detail below, will further increase my enthusiasm for the current study.

1. It is not clear how performing meta-analysis across multiple population groups aids the analysis compared to only using the European group. Actually, it might be true that the multi-ancestry analysis does not enhance the analysis at all. I suggest that the authors mention how many more significant vQTLs are identified from a cross-population meta-analysis compared to a European-only analysis, and comment on their findings. As currently presented, the incorporation of multi-ancestry seems like a skipped over point.

One suggestion: since participants may lie in a continuum of ancestry, rather than in distinct population groups, could the authors have performed a joint analysis instead of a meta-analysis? This would also help avoid some of the issues with Levene's test in data sets with few individuals. As an example, Wojcik et al. Nature 2019 used SUGEN and GENESIS for GWAS, and suggested this would improve power and maintain false positive rate over a meta analysis approach.

2. The authors have a distinctive opportunity to leverage multiple cohorts in vQTL based analyses. Currently, WGHS is only used for replication, but it would be interesting to further incorporate WGHS in the analyses.

a) First, could environmental exposures (which are not present in UKB) be tested for GEI in WGHS? This would be a neat approach, showing that vQTLs can be first identified in UKB and then followed up in more specialized cohorts such as WGHS.

b) Secondly, would it be possible to use WGHS in a meta-analysis framework to improve discovery of vQTLs? This could lead to additional SNPs in the stage 2 analysis.

3. I believe the fact that 8.4% of vQTLs are not ME loci is a large number, and an analysis plus discussion of this should be included.

- Where are these 8.4% vQTLs located? Are many in the MHC?
- Are ME effects present (e.g. $P < 0.05$), but just not detected at genome-wide significance? For these loci, do the ME effects generally correlate in the same direction as the vQTL effects?
- In GxE testing, are they less likely to detect an interaction compared to those with a ME effect? This

could be a measure of a spurious vQTL association.

- At the non-ME vQTLs w/ GxE interactions, do the SNP effects conditional on the environmental exposure switch direction? As an example, the ADH1B-alcohol interaction is one of these, where never has an association of increased ALT with number of T alleles, but at least once leads to association of decreased ALT with number of T alleles. This could lead to a variance signature but not a mean association.

- What distinguishes vQTLs from ME loci? Are there any interesting biological properties that distinguish vQTLs from ME loci? For example, the authors could contrast the epigenomic information of vQTLs to compare the gene regulation of vQTLs. Or, the gene sets of vQTLs and ME loci.

4. The authors identified 846 GEIs. However, many of these may be driven by the same overall environmental exposure, since there are so many environmental exposures measured and many of these exposures are very similar. To identify independent GEIs, can the authors repeat the analysis using a joint model of multiple interactions? Say, include all significant interactions from stage 2 into a new model which includes all interactions simultaneously. This will help pinpoint the true GEIs.

Other comments:

5.

a)

"Genetic correlation magnitudes tended to be similar (mean $|\rho_g|$ of 0.16 across all non-identical biomarker pairs), while $\rho_{g,ME}$ p-values were substantially lower"

I think this needs clarification. Is this comparing genetic correlation of mean GWAS versus genetic correlation of variance GWAS? If so, what is "mean $|\rho_g|$ "?

b) Also:

Should $|\rho_{g,vQTL}| - |\rho_{g,ME}| > 0.2$ be rewritten as $|\rho_{g,vQTL} - \rho_{g,ME}| > 0.2$?

6.

"Its GEI effect with BMI represents a small but significant increase in the positive association between BMI and serum TG ($p = 5.05 \times 10^{-25}$)."

Does the p value refer to the interaction p value?

7.

"inverse-normal transformed phenotypes, which have no mean-variance relationship, produce largely similar vQTL results" -The inverse normal transformation does not completely remove the mean-variance relationship. See Young et al. Nature Genetics 2018 on the HLMM method, which is needed to identify SNP associations with phenotypic variability independent of mean effects, and Marderstein et al. AJHG 2021 for further discussion. Thus, it should be expected that vQTLs in inverse-normal transformed phenotypes will largely recapitulate vQTLs prior to transformation.

8.

The authors state "the presence of a vQTL further increases the likelihood of identifying an underlying interaction". However, the authors did not prove this in their own data by also performing GEI analysis on the ME loci or a control set of SNPs, and comparing GEI discovery rates to the vQTLs. The authors should either clearly state that this was already shown in prior literature (e.g. Wang et al Science Advances 2019 and Marderstein et al AJHG 2021) and would likely extend to the current analysis, or better yet perform an analysis of their own.

9.

"Of note, cholesterol, LDL-C, and Apolipoprotein B were also adjusted for statin use using methods described previously" For clarity, the authors should briefly describe methods for how they adjusted these measures for statin use.

10. I did have 3 minor issues with the figures presented within the paper, and have outlined them below.

a)

I think the Figure 1 caption should be elaborated briefly to describe the schematic figure.

b)

The Figure 2 text is too small, and thus difficult to read.

c)

Figure 5b would work better as a table. Many of the points are not labeled, and thus it is unclear how to best understand and interpret the unlabeled points. Furthermore, it is unclear how this plot would look for other SNPs, such as if the number of GEI (at $P < 0.05$) is more or less compared to the number at other SNPs.

Reviewer #2 (Remarks to the Author):

Westerman et al. conducted a two-stage GEI detection study, i.e. variance-quantitative trait loci (vQTL) + exposure-wide interaction study (EWIS), for 20 cardiometabolic biomarkers with common ($MAF > 0.005$) imputed genotype and 2,380 PHEASANT-generated exposures in the UKB, and replicated vQTL and GEI results in an independent dataset WGHS. Two GEIs were highlighted: 9 variants interacted with body mass/body fat on triglyceride, and ADH1B locus interacted with alcohol intake on liver stress marker ALT.

To me, there are two novelty: 1) applied vQTL analysis on cardiometabolic biomarkers; 2) systematically scanned GEI with a large number (2,380) of exposures. The paper is well-structured and clearly written and I have a few major and minor issues.

Major:

1) log-transformation: Wang et al. (PMID: 31453325, Figure 2b) showed many non-linear transformations, including log-transformation, could inflate the test-statistic of Levene's test on vQTL detection, while the main vQTL analysis of this paper used log₂-transformed phenotype. Will this log₂-transformation lead some false positive findings?

2) The ancestry-specific vQTL analysis is interesting and new to me. The authors said "the non-European findings are the result of spurious association given lower sample sizes, especially at lower minor allele frequencies", which need further explanation or exploration. Are the vQTLs specific for non-European populations with lower MAFs? A scatter plot with three genotype groups and phenotype values may help (similar to Figure S3a in Wang et al.).

3) The authors applied some methods originally developed for main-effect GWAS test statistics on vQTL test statistics, like multi-ancestry meta-analysis based on Levene's test statistics derived effect size and LDSC. However, the underlying assumptions of these methods may not hold from main effect to variance effect. For example, the derivation/back-transformation from p value and test statistics to effect size and standard error based on Zhu et al. (PMID 27019110, supplementary note 2) was based on simple linear regression, which may be not suitable or to be justified for Levene's test. Note that while the OSCA software provided the effect size and standard error in its output file (<https://yanglab.westlake.edu.cn/software/osca/#vQTLAnalysis>), Wang et al. (PMID: 31453325) did not perform any downstream analysis based on these values. Also did multi-ancestry meta-analysis assume "the effect size is fixed across different ancestry"? How reliable is it for variance or GEI effects? And what LD score dataset should be used in LDSC analysis for multi-ancestry meta-analysis summary data? Did the authors just used that based on European individuals provided by Alkes group (<https://github.com/bulik/ldsc/wiki/LD-Score-Estimation-Tutorial>)?

Minor:

- 1) More detailed information should be provided for others to understand or replicate this work, which includes:
 - a. The exact number of (common) genetic variants tested;
 - b. The list of these 2,380 exposures (not only summarised statistics in Table S7);
 - c. For the online portal, ancestry-specific summary data, and downloadable text files containing all the variants (not only $p < 0.01$);
- 2) Introduction: a few more sentences to introduce the GEI research in the cardiometabolic/cardiovascular field.
- 3) Replication study in WGHS: for 61 vQTLs replicated with a nominal threshold ($p < 0.05$), how many have the same direction?
- 4) Two GEI examples in Results section: have "rs139566989" in LIPC gene and "rs1229984" in ADH1B gene been proved to be the causal variants, while LIPC and ADH1B are likely to be causal genes due to its relevant biological function? The two SNPs could also be tagging the causal variants through LD.
- 5) 7th paragraph in Discussions (last paragraph in page 7): "they can nonetheless can play a role". The word "can" appears twice.
- 6) 2nd paragraph in the "UK Biobank genetic data" subsection of Methods section, it should be "African" instead of "West African", as I think Kenya (LWK subpopulation) is an East African country.
- 7) The same paragraph: explain more or provide a reference for "probabilistic Gaussian mixture models were built to represent normally-distributed subpopulations ..." and "Rand index score".
- 8) Figure 4 and 5: indicate the genome-wide or study-wide thresholds. Say, a vertical dash line indicated the genome-wide significant threshold in Figure 5a.
- 9) The legend of Figure S1, it said the phenotype processing consisted "winsorization", while in the "serum biomarkers" subsection of Methods section, it said "outliers were set to missing". I think these two processes are different (<https://en.wikipedia.org/wiki/Winsorizing>).

Manuscript number: NCOMMS-21-45458-T

Title: Variance-quantitative trait loci enable systematic discovery of gene-environment interactions for cardiometabolic serum biomarkers

Thank you to the reviewers for your effort and time in reviewing our manuscript. In response to these comments, we have made a few major changes that we highlight here:

- We replaced the previous beta/SE-based meta-analysis method (for cross-ancestry vQTL meta-analysis) with one based on Stouffer's method, which is more appropriate for statistical tests such as Levene's test that have no inherent effect estimate. Changes to the results were minor (for example, of 182 prior vQTL loci, 4 additional loci were gained and 2 were lost). However, the revised methodology required propagation of new results throughout the computational pipeline, including the subsequent exposome-wide interaction study.
- We have implemented an iterative conditional interaction analysis to allow us to determine the number of *independent* GEIs (accounting for the strong inter-exposure correlation), which is now reported in the Abstract and Results.
- We have devoted a new paragraph to a deeper investigation of the vQTL loci lacking a genetic main effect, in terms of their variant type, enrichment for underlying GEIs, and statistical considerations.

Below we outline our responses to the reviewer's specific comments in blue.

RESPONSE TO REVIEWERS' COMMENTS

Reviewer #1 (Remarks to the Author):

“Variance-quantitative trait loci enable systematic discovery of gene-environment interactions for cardiometabolic serum biomarkers” is a very interesting and well-written manuscript. Identifying gene environment interactions is difficult, and I am impressed by the number of interactions discovered by employing a two-stage, variance QTL based approach. The authors analyze their results to present several unique findings, such as teasing out triglyceride interactions with either body mass or body fat exposures, and a strong ADH1B-alcohol interaction despite no mean-based association at the ADH1B locus.

Overall, I am impressed by the sheer number of associations and several unique findings. While the technical analyses appear to be robust and valid, I would like to see the work strengthened by elaborating on their analyses and results. Particularly:

(1) it is not clear how multi-ancestry aids the analysis,

- (2) I would like to see if the multi-cohort context could be further leveraged in vQTL analysis,
- (3) it is not clear what contrasts vQTLs from ME loci, and
- (4) how many independent interactions exist.

Addressing these points, which are described in greater detail below, will further increase my enthusiasm for the current study.

1. It is not clear how performing meta-analysis across multiple population groups aids the analysis compared to only using the European group. Actually, it might be true that the multi-ancestry analysis does not enhance the analysis at all. I suggest that the authors mention how many more significant vQTLs are identified from a cross-population meta-analysis compared to a European-only analysis, and comment on their findings. As currently presented, the incorporation of multi-ancestry seems like a skipped over point.

One suggestion: since participants may lie in a continuum of ancestry, rather than in distinct population groups, could the authors have performed a joint analysis instead of a meta-analysis? This would also help avoid some of the issues with Levene's test in data sets with few individuals. As an example, Wojcik et al. Nature 2019 used SUGEN and GENESIS for GWAS, and suggested this would improve power and maintain false positive rate over a meta analysis approach.

The reviewer brings up an important point about the maximization of sample size and associated power gain via a pooled analysis rather than an ancestry-specific meta-analysis. We chose our ancestry-specific analysis plus meta-analysis strategy for two major reasons. First, it avoids issues of ancestry-specific variance heterogeneity (whether due to genetic effects or cultural/ethnicity-related behavioral differences) that could otherwise result in false positives driven by population stratification. This can be dealt with for GWAS using ancestry-specific variances such as in SUGEN and GENESIS as suggested by the reviewer, but is more complex in a variance-focused analysis like this one and we are not aware of an analytical approach that allows for this heterogeneity while performing vQTL tests. Interpretation of underlying interactions would also be complicated by the association between ancestry groups and exposures driven by cultural and behavioral differences. Second, our stratified approach allows us to develop ancestry-specific summary statistics and thus identify population-specific variance effects, as currently reported in Supp. Table S4 and discussed in response to Reviewer #2 point 2 below.

In terms of increased discovery via including other ancestries compared to the EUR subset alone, we find five loci that pass a Bonferroni threshold in the meta-analysis but not in the EUR subset. We now mention this effect explicitly in the Results by modifying an existing sentence: "...P-values from the vQTL meta-analysis tracked closely with those from the European subset (Supp. Fig. S2), while uncovering five additional loci that were not significant in the European subset alone."

Inclusion of admixed individuals would present similar difficulties in confounding and interpretation as an ancestry-pooled analysis. In terms of the potential power gain offered by the inclusion of these individuals, we note that there are a total of ~407,000 participants used in the central UKB principal components analysis process (unrelated high quality samples; see Bycroft et al. 2018), compared to

the ~395,000 that were assigned exclusively to one of the four ancestry groups in our analysis. So, inclusion of admixed individuals here would result in the inclusion of only an additional ~12,000 individuals. This increase of 3% in sample size is modest compared to the current analysis sample, but could nonetheless improve representation of diversity in both genetic ancestry and culturally-driven exposures, so is an approach that we plan to pursue in the future. See: Bycroft, C. *et al.* The UK Biobank resource with deep phenotyping and genomic data. *Nature* **562**, 203–209 (2018).

To further explore the impact of pooling and whether we might have any notable power loss due to the use of meta-analysis, we chose ALT as a representative biomarker and conducted an analogous genome-wide vQTL analysis using the pooled group of all four ancestry groups. The only differences compared to our primary ALT analysis were: (1) no need for the follow-up meta-analysis, and (2) adjustment for UKB-wide principal components (retrieved from the Pan-UKBB analysis effort: <https://pan.ukbb.broadinstitute.org/>) in place of ancestry-specific principal components. The use of UKB-wide principal components in this analysis allowed us to adjust at least partially for potential ancestry-driven effects. The plot below shows that results from this pooled approach and the meta-analysis approach were largely similar, especially for the most significant vQTL hits. We note that this test only explores the potential effect of pooling, but not the potential power gain from the additional ~12k admixed individuals.

Comparison of $-\log_{10}(p)$ for ALT based on vQTL meta-analysis vs. pooled analysis:

Having explored this topic in substantially greater depth in response to the reviewer's suggestion, we do think it is a potentially beneficial approach and hope to incorporate it in future similar analyses. However, we believe it deserves more direct attention and exploration (for example, through simulation studies) than is appropriate for the current study.

2. The authors have a distinctive opportunity to leverage multiple cohorts in vQTL based analyses. Currently, WGHS is only used for replication, but it would be interesting to further incorporate WGHS in the analyses.

a) First, could environmental exposures (which are not present in UKB) be tested for GEI in WGHS? This would be a neat approach, showing that vQTLs can be first identified in UKB and then followed up in more specialized cohorts such as WGHS.

This is a great suggestion by the reviewer to demonstrate additional value of vQTL discovery in biobanks like UKB by finding GEIs in additional cohorts with more extensive phenotyping in some domain. We pursued this by exploring the available phenotypes in WGHS, but found that few additional exposure traits were available that were not already included in the exposome-wide set of phenotypes in UKB. In future work, we would like to pursue this question in cohorts with more detailed environmental phenotypes and longitudinal data.

b) Secondly, would it be possible to use WGHS in a meta-analysis framework to improve discovery of vQTLs? This could lead to additional SNPs in the stage 2 analysis.

While the incorporation of additional cohorts would be of great value for vQTL discovery, this is outside of our intended scope for this investigation. WGHS would not make a major difference in the overall sample size for discovery (~5% difference), and has only half of the relevant biomarkers available for meta-analysis. As conducted, we are able to provide a more homogenous set of UKB-based summary statistics that is applied consistently across biomarkers. Therefore, by keeping this first analysis focused on a single cohort, we were able to maintain power while not needing to consider how study-specific factors could influence the analysis and interpretation of the results, and also ensuring that the analysis across biomarkers is consistent. We hope to build on this approach in future investigations, bringing in additional cohorts and potentially incorporating a pooled analysis framework as suggested above in point #1.

3. I believe the fact that 8.4% of vQTLs are not ME loci is a large number, and an analysis plus discussion of this should be included.

Overall, there are 17 variant-biomarker pairs (16 unique variants) that have a significant vQTL but a non-significant ME ($p < 4.51 \times 10^{-9}$). The reviewer makes a set of excellent suggestions on how to further understand and communicate the relevance of these loci. The text immediately below has been added to the Results, incorporating what we feel are the most meaningful findings. Responses to the specific suggestions are also provided below.

“The 17 vQTL-biomarker pairs without corresponding main effects deserve specific consideration. First, we note that they have a wide distribution of main effect p-values (including eight with $p > 0.05$) rather than clustering just above the Bonferroni significance threshold. Thus, some of these loci may truly lack genetic main effects, especially given the greater power of standard linear regression compared to Levene’s test (Pare...Chasman 2010). Upon deeper investigation, we note that these 17 vQTLs are evenly

spread across both chromosomes and biomarkers. While there are more vQTLs with at least one GEI among those that also have a ME (28%) compared to a vQTL effect only (18%), this difference is not significant (Chi-square test $p = 0.55$) and does not provide evidence that vQTLs without a ME represent spurious associations. Notably, to induce a vQTL effect in the absence of a significant ME, genotypic effects on phenotype may have opposite signs in some subset of the population. In some instances, this may be explained by an underlying qualitative interaction (see the alcohol-*ADH1B* example below), while in other instances the relevant stratifying trait may remain unknown. Thus, rather than being spurious, at least some subset of these 17 vQTLs appears to truly lack overall MEs.”

- Where are these 8.4% vQTLs located? Are many in the MHC?

After looking into this set of loci (vQTLs lacking overlapping MEs) more in-depth, they are fairly evenly spread across chromosomes (with at least one signal in each of 10 chromosomes) and across biomarkers (in similar proportions to the amount of vQTLs per biomarker).

- Are ME effects present (e.g. $P < 0.05$), but just not detected at genome-wide significance? For these loci, do the ME effects generally correlate in the same direction as the vQTL effects?

Our existing Supplemental Figure S3a (updated to explicitly show all non-significant ME p-values and pasted below) aims to assess the relationship between vQTL and ME significance. Eight of the seventeen vQTLs without overlapping MEs have suggestive ME p-values ($0.05 > p >$ Bonferroni threshold; also see plot below). Of these nine, the effect directions are consistent for seven variants (acknowledging the limitations in assigning clear directions of effect based on Levene’s test).

- In GxE testing, are they less likely to detect an interaction compared to those with a ME effect? This could be a measure of a spurious vQTL association.

To address this, we tested for an enrichment of underlying GEIs among vQTLs with and without a corresponding ME. The following table counts locus-biomarker pairs with or without an associated ME (rows) and with or without at least one significant underlying GEI at that locus in the downstream EWIS

(columns). Eighteen percent (3/17) of vQTLs lacking MEs had a significant GEI versus 28% (46/167) of vQTLs with corresponding MEs. While it does appear that a vQTL with a ME is more likely to have at least one underlying GEI, the difference is not statistically significant (Chi-square test $p = 0.55$). Based on these results, there isn't strong evidence that the vQTLs lacking corresponding MEs are spurious associations.

	0 GEI	≥ 1 GEI	Fraction with ≥ 1 GEI
No corresponding ME	14	3	18%
Corresponding ME	121	46	28%

- At the non-ME vQTLs w/ GxE interactions, do the SNP effects conditional on the environmental exposure switch direction? As an example, the *ADH1B*-alcohol interaction is one of these, where never has an association of increased ALT with number of T alleles, but at least once leads to association of decreased ALT with number of T alleles. This could lead to a variance signature but not a mean association.

The alcohol-*ADH1B* example is a great example of the way vQTLs can arise without an associated main effect. As the reviewer suggests, SNP rs1229984 associates positively with ALT in the “never/special occasions” alcohol group, modestly negatively in the “<1 per day” group, and substantially negatively in the “at least once per day” group (see table below).

Indeed, we can see this pattern for other such variants. For example, the strongest EWIS interaction involving one of the non-ME vQTL variants is between rs738408 and waist circumference (WC) influencing triglycerides (TG). When stratifying into tertiles of waist circumference, we see the sign of the SNP-TG association change sign:

WC tertile	SNP estimate	P-value
T1	-0.02801589	2.981912e-10
T2	0.00647596	2.104763e-01
T3	0.04134322	4.905931e-14

We have now referenced this phenomenon as a general discussion (see new Results text above in response to the overall point #3) and reinforced it specifically in the section about the alcohol vignette: “This interaction also demonstrates how a strong vQTL effect can arise with a minimal overall ME: rs1229984 is associated positively with ALT in the never or rare alcohol consumers, but negatively in frequent consumers (Fig. 5c), thus ‘canceling each other out’ in the marginal ME test.”

- What distinguishes vQTLs from ME loci? Are there any interesting biological properties that distinguish vQTLs from ME loci? For example, the authors could contrast the epigenomic information of vQTLs to compare the gene regulation of vQTLs. Or, the gene sets of vQTLs and ME loci.

We investigated this question using the LOLA program to understand whether vQTLs or MEs have differential enrichments for epigenomic annotations. LOLA calculates enrichment of a set of target regions of the genome for enrichment in a set of epigenomic annotations compared to a background “universe” set of loci. By choosing this background to be the union of the vQTL and ME loci, the resulting significance tests indicate a direct comparative enrichment between the two groups for any given epigenomic annotation. Upon conducting this set of enrichment tests across the LOLA “core” set of epigenomic annotations (ENCODE TFBS, ENCODE genome segmentation, Cistrome, Sheffield 2013 DNase hypersensitivity sites, UCSC features [e.g., CpG islands], and the CODEX database), we found no significant comparative enrichments at false discovery rate of 0.05 based on the Benjamini-Hochberg method.

We briefly describe this analysis in the Results: “vQTL and ME loci did not show significant differential enrichment for epigenomic features using the Locus Overlap Analysis (LOLA) method (see Methods).”

And the Methods:

“Differential enrichment for epigenomic features between vQTL and ME loci was assessed using the Locus Overlap Analysis (LOLA) strategy and associated R package. LOLA calculates enrichment of a set of target regions of the genome for enrichment in a set of epigenomic annotations compared to a background “universe” set of loci. The basic “LOLA core” set of epigenomic annotation collections was used. The background locus set was chosen to be the union of the vQTL and ME loci such that the resulting significance tests directly compared enrichment between the two groups for any given epigenomic annotation. Significance was assessed after application of a Benjamini-Hochberg false discovery rate correction.”

4. The authors identified 846 GEIs. However, many of these may be driven by the same overall environmental exposure, since there are so many environmental exposures measured and many of these exposures are very similar. To identify independent GEIs, can the authors repeat the analysis using a joint model of multiple interactions? Say, include all significant interactions from stage 2 into a new model which includes all interactions simultaneously. This will help pinpoint the true GEIs.

We agree that it would be valuable to get a better idea of the number of effective or independent GEIs, and appreciate the reviewer’s suggestion on how to approach this. We are concerned about the substantial collinearity involved with a joint model containing all significant exposure interactions simultaneously. However, we can adapt the iterative conditional analysis approach that is standard for uncovering secondary, independent signals at GWAS loci. Here, instead of conditioning on the top SNP at each locus, we take a given vQTL-biomarker pair, and successively condition on the most-significant exposure GEI until there are no remaining signals reaching significance (either experiment-wide or nominal). Taking this approach for all of the 49 significant vQTL-biomarker pairs, the 847 “raw” GEIs are reduced to 66 independent GEIs (at the stringent experiment-wide p-value threshold) or 132 independent GEIs (at a nominal threshold of $p < 0.05$).

We have reported this in the Results:

“Many of the participating exposures were from highly correlated categories such as anthropometric measures. To address this, we iteratively conditioned GEI tests on the most significant exposure interaction at each locus to estimate the number of independent GEIs. We found 66 independent GEIs when retaining the same strict Bonferroni threshold for “secondary signals”, which increased to 132 GEIs when using a nominal threshold for these secondary signals.”

As well as in the Methods:

“Because of the substantial correlation between many tested exposures, we conducted a follow-up analysis to determine the number of independent GEIs. For each vQTL-biomarker pair participating in a significant GEI from the primary EWIS (and thus significant at the strict Bonferroni threshold), we re-tested for GEI with each exposure. Then, we iteratively re-tested while additionally conditioning on the GEI effect for the most-significant exposure. We report the number of total independent GEIs, counting secondary signals using both a nominal significance threshold ($P < 0.05$) and using our stricter EWIS Bonferroni threshold.”

And finally in the Abstract:

“Next, we tested each vQTL for interaction across 2,380 exposures, identifying 847 significant GEIs ($p < 2.4 \times 10^{-7}$), of which 132 were independent ($p < 0.05$) after accounting for correlation between exposures.”

Other comments:

5.

a)

“Genetic correlation magnitudes tended to be similar (mean $|r_g|$ of 0.16 across all non-identical biomarker pairs), while $r_{g,ME}$ p-values were substantially lower”

I think this needs clarification. Is this comparing genetic correlation of mean GWAS versus genetic correlation of variance GWAS? If so, what is “mean $|r_g|$ ”?

The mean $|r_g|$ value across all non-identical biomarker pairs is the same (0.16) whether that mean $|r_g|$ comes from vQTL or ME results. We intended to communicate that, for each of vQTL-based and ME-based genetic correlations, the *magnitudes* of genetic correlation tended to be similar whereas the *significances* were notably stronger for ME-based genetic correlations. We attribute this to the larger standard errors (explored by Marderstein et al. in Supp. Note S4). This was not clear in the original text, so we have updated the sentence:

“We observed similar genetic correlations between biomarkers when using vQTL ($r_{g,vQTL}$) or main-effect ($r_{g,ME}$) summary statistics (Spearman correlation of 0.69 between $r_{g,vQTL}$ and $r_{g,ME}$ values across all non-identical biomarker pairs; Supp. Fig. S4). The mean $|r_{g,vQTL}|$ and mean $|r_{g,ME}|$ were also equal (0.16); therefore, while the associated significances of $r_{g,ME}$ tended to be substantially stronger,

likely due to the known larger standard errors in variance estimation compared to mean estimation¹¹, the genetic correlation magnitudes tended to be similar.”

b) Also:

Should $|r_{g,vQTL} - r_{g,ME}| > 0.2$ be rewritten as $|r_{g,vQTL} - r_{g,ME}| > 0.2$?

We initially did intend to compare magnitudes (i.e., absolute values from each r_g type) to avoid situations in which non-significant estimates with opposite signs crossed zero and resulted in a seemingly meaningful difference of >0.2 . Upon considering the reviewer’s point, we updated this to instead consider the magnitude of raw differences ($|r_{g,vQTL} - r_{g,ME}|$) while also requiring at least nominal significance for both r_g estimates. This produced three biomarker-biomarker pairs with substantially different magnitudes, two of which had greater magnitudes for $r_{g,vQTL}$. We have updated the text to reflect this:

“For three biomarker pairs, the $r_{g,vQTL}$ estimate was substantially different ($|r_{g,vQTL} - r_{g,ME}| > 0.2$ with nominal significance for both estimates. For example, one of these involved a greater $r_{g,vQTL}$ between HbA1c and ALT; these biomarkers may thus be more similar in their modifiable genetic effects (through GEIs, for example) than their fixed genetic effects.”

6.

“Its GEI effect with BMI represents a small but significant increase in the positive association between BMI and serum TG ($p = 5.05 \times 10^{-25}$).” Does the p value refer to the interaction p value?

This number is intended to reflect the interaction p-value, but mistakenly the vQTL, rather than GxE, p-value had been provided. The sentence has been revised as follows:

“Its GEI effect with BMI represents a small but significant increase in the positive association between BMI and serum TG (p -interaction = 6.05×10^{-9}).”

7.

“inverse-normal transformed phenotypes, which have no mean-variance relationship, produce largely similar vQTL results” –The inverse normal transformation does not completely remove the mean-variance relationship. See Young et al. Nature Genetics 2018 on the HLMM method, which is needed to identify SNP associations with phenotypic variability independent of mean effects, and Marderstein et al. AJHG 2021 for further discussion. Thus, it should be expected that vQTLs in inverse-normal transformed phenotypes will largely recapitulate vQTLs prior to transformation.

This is a good point about the implications of the inverse-normal transformation on vQTL detection. While the INT procedure will tend to reduce the correlation between main effect and variance effects, the reviewer rightly points out that the INT does not in general remove this correlation altogether (ex. from Young et al.: “variance effects due to a mean–variance relationship are not, in general, removed by inverse normal transformation.”).

We have corrected this sentence in the text: “...our sensitivity analyses demonstrate that inverse-normal transformed phenotypes, for which the correlation between mean and variance effects is reduced, produce largely similar vQTL results.”

With this said, we don't believe that vQTLs from INT phenotypes will trivially recapitulate most "raw" vQTLs. Empirically, it appears that vQTL results from raw or INT phenotypes do not strongly overlap. For example, Marderstein et al. tested for BMI vQTLs genome-wide both before ("raw") and after ("INT") INT transformation. They found limited overlap between the two (14 raw-only, 20 INT-only, and only 7 shared; calculated from their Fig. 3H).

Ultimately, we cannot use our sensitivity tests based on INT phenotypes to guarantee that associated vQTLs are independent of the mean-variance relationship. However, we do believe that the sensitivity analyses using INT traits provide an additional degree of confidence that variants having ME and vQTL effects are not purely artifacts. This point also contributes to our choice to log-transform all biomarker phenotypes, which generally produced more normal distributions and thus avoided a greater number of false positives due to the mean-variance phenomenon (further Discussion in response to Reviewer 2, point #1 below).

8.

The authors state "the presence of a vQTL further increases the likelihood of identifying an underlying interaction". However, the authors did not prove this in their own data by also performing GEI analysis on the ME loci or a control set of SNPs, and comparing GEI discovery rates to the vQTLs. The authors should either clearly state that this was already shown in prior literature (e.g. Wang et al Science Advances 2019 and Marderstein et al AJHG 2021) and would likely extend to the current analysis, or better yet perform an analysis of their own.

Based on the reviewer's suggestion, we ran the exposome-wide interaction analysis on the set of ME variants from Stage 1 of our pipeline. Given our focus on vQTL prioritization, we believe it is beyond the scope of this manuscript to report exhaustively on the GEI findings from this analysis, but we can use these results to test for enrichment of significant GEIs at vQTL-biomarker pairs compared to ME-biomarker pairs. Testing the 4413 variant-biomarker pairs with significant ME effects for interaction with each of the 2380 exposures, we found the following:

- Of 437,920 vQTL-biomarker-exposure triplets, 847 (0.20%) have a significant GEI. Of 10,502,940 ME-biomarker-exposure triplets, 1,590 (0.02%) have a significant GEI. This substantially greater enrichment of GEI signal for vQTL variants is highly significant (Chi-square test $p < 1e-16$).
- At the level of variant-biomarker pairs, vQTLs are also more enriched for GEI signal. 49 out of 184 vQTL-biomarker pairs (26.6%) have at least one significant GEI, compared to only 145 out of 4,413 ME-biomarker pairs (3.3%) (Chi-square test $p < 1e-16$).

These results clearly support the enrichment of vQTLs for underlying GEIs compared to MEs, matching previous conclusions using different phenotypes from the articles mentioned by the reviewer (and already cited in our manuscript in support of this point). Furthermore, we note that this analysis uses the same EWIS Bonferroni threshold for both groups of loci (vQTLs & MEs). If using MEs in practice, the greater number of loci and stricter Bonferroni threshold would decrease the number of GEIs uncovered and further increase the enrichment difference.

We have added the following to the Results:

“We also conducted a full EWIS on the larger set of 4,413 ME-biomarker pairs in order to compare enrichment of GEI findings with those from vQTLs. Replicating findings from previous studies (cite Wang and Marderstein), we saw a substantial enrichment for GEIs within vQTL-biomarker pairs (26.6% of vQTLs had a significant GEI compared to 3.3% for MEs; $p = 1.8e-52$).”

And the Methods:

“To compare enrichment of vQTL versus ME loci for underlying GEIs, we conducted a parallel EWIS using the larger set of ME loci. Other than the different set of input variants (ME index variants rather than vQTL index variants), this analysis was identical to the primary EWIS (including the use of the same Bonferroni threshold).”

9.

“Of note, cholesterol, LDL-C, and Apolipoprotein B were also adjusted for statin use using methods described previously” For clarity, the authors should briefly describe methods for how they adjusted these measures for statin use.

We have added an additional sentence after the one in question with further details:

“Briefly, in individuals with self-reported use of a statin medication, each of these biomarkers was divided by an adjustment factor (0.749, 0.684, and 0.719, respectively) that had been empirically estimated by Sinnott-Armstrong and colleagues in the same population.”

10. I did have 3 minor issues with the figures presented within the paper, and have outlined them below.

a)

I think the Figure 1 caption should be elaborated briefly to describe the schematic figure.

The caption is now expanded from simply “Analysis workflow” to the following:

“**Figure 1:** Analysis workflow. Unrelated individuals without major disease from any of four ancestry groups in the UK Biobank were included in the analysis. Twenty metabolic biomarkers were preprocessed, including log-transformation, adjustment for biological and technical covariates, and outlier removal. vQTL analysis was performed genome-wide, separately in each ancestry group followed by multi-ancestry meta-analysis. Significant variant-biomarker pairs were taken forward for GEI analysis, along with 2380 exposures preprocessed using the PHESANT tool. Finally, GEI tests were performed for each combination of variant-biomarker-exposure triplet, again using ancestry-stratified analysis followed by meta-analysis.”

b)

The Figure 2 text is too small, and thus difficult to read.

We have updated the figure to have larger text in proportion to the plot sizes.

c)

Figure 5b would work better as a table. Many of the points are not labeled, and thus it is unclear how to best understand and interpret the unlabeled points. Furthermore, it is unclear how this plot would look for other SNPs, such as if the number of GEI (at $P < 0.05$) is more or less compared to the number at other SNPs.

Figure 5 is a close-up exploration of our *ADH1B* locus, which accompanies the section labeled “*Alcohol intake interacts with a common ADH1B polymorphism to influence liver stress*”. Figure 5b highlights the specificity of the interaction to alcohol consumption, and is not meant to detail the number or identity of the other nominal GEIs. We agree this is important information for readers to access; all nominally-significant GEIs are available for viewing and downloading on our data portal (<https://hugeamp.org/research.html?pageid=UKB-vQTL-GxE>).

Reviewer #2 (Remarks to the Author):

Westerman et al. conducted a two-stage GEI detection study, i.e. variance-quantitative trait loci (vQTL) + exposure-wide interaction study (EWIS), for 20 cardiometabolic biomarkers with common ($MAF > 0.005$) imputed genotype and 2,380 PHEASANT-generated exposures in the UKB, and replicated vQTL and GEI results in an independent dataset WGHS. Two GEIs were highlighted: 9 variants interacted with body mass/body fat on triglyceride, and *ADH1B* locus interacted with alcohol intake on liver stress marker ALT.

To me, there are two novelty: 1) applied vQTL analysis on cardiometabolic biomarkers; 2) systematically scanned GEI with a large number (2,380) of exposures. The paper is well-structured and clearly written and I have a few major and minor issues.

Major:

1) log-transformation: Wang et al. (PMID: 31453325, Figure 2b) showed many non-linear transformations, including log-transformation, could inflate the test-statistic of Levene’s test on vQTL detection, while the main vQTL analysis of this paper used log₂-transformed phenotype. Will this log₂-transformation lead some false positive findings?

We considered this finding from Wang et al. (and replicated in simulations from Marderstein et al.) when deciding on our final analytic approach, but ultimately decided to use log-transformed phenotypes for multiple reasons.

First, we wanted to keep consistency with an existing main-effect GWAS of biomarkers in the UK Biobank (from Sinnott-Armstrong et al., cited in our manuscript), for comparison and to build on the substantial amount of effort that this paper put into quality control. This phenotyping process ultimately log-transformed all biomarkers prior to analysis.

Second, the simulations from Wang et al. and Marderstein et al. were conducted using generative models on the raw scale, such that the true variance heterogeneity and genetic main effect, etc. arose with respect to the raw phenotype. They then showed inflated false positives when testing on a different (log-transformed) scale. Because our analysis proceeds from start to finish using log-transformed phenotypes

(in both Stage 1 and 2), we are instead seeking variance heterogeneity and subsequently GEIs where both the quantity of interest and the analysis pertain to a log-transformed phenotype.

Third, we conducted pilot analyses using both raw and log-transformed phenotypes to compare results. We found clear false positives for many biomarkers when using non-log-transformed phenotypes. While the lambda values for all biomarkers between the raw and log-transformed phenotypes were similar across all ancestries (raw interquartile range: 1.031-1.105 and log-transformed interquartile range: 1.048-1.097), there was clearly a large amount of inflation in the vQTL Manhattan plots for the raw phenotypes, particularly in the non-European ancestry analyses. For example, the Manhattan plots for the ALT liver enzyme levels vQTL analyses across all 4 ancestry groups using raw and log-transformed phenotypes are shown below. Note, these plots are all limited to $P < 0.01$, and also $P > 1e-16$ for EUR.

We have clarified these points in the Methods section: “Log-transformation was chosen because it most closely mirrored the GWAS by Sinnott-Armstrong et al. and because initial empirical results showed substantial inflation of vQTL p-values for highly-skewed raw biomarker values.”

2) The ancestry-specific vQTL analysis is interesting and new to me. The authors said “the non-European findings are the result of spurious association given lower sample sizes, especially at lower minor allele frequencies”, which need further explanation or exploration. Are the vQTLs specific for non-European

populations with lower MAFs? A scatter plot with three genotype groups and phenotype values may help (similar to Figure S3a in Wang et al.).

Though we believe it's important to include analyses and provide results for non-European populations, we were hesitant to put substantial emphasis on the ancestry-specific results due to the possibility for spurious results (exemplified in Fig. S3a from Wang et al.), given the smaller sample sizes in non-Europeans and a more liberal MAF filter (1%) as compared to Wang et al (5%).

The reviewer's suggestion to visualize non-European vQTLs is a good one. It allows us to assess spurious variance effects due to a single or small number of outlier values in a low-frequency genotype, as was done in Wang et al. Fig. S3a (copied below; top plot). Specifically, visualization by genotype allows the potential detection of a single or small number of outlier values in a low-frequency genotype (genotype 2 in the plot) that spuriously inflate the variance for that genotype.

Plot from Fig. S3a from Wang et al.:

After plotting two of our most significant non-European vQTLs (pair of plots below), it does appear that higher variance in the very low-allele count groups could be driving false-positive variance signals.

Indeed, applying median-based Levene's test in the associated discovery ancestry group supports the potential for false-positive association at low allele frequency. For example, for the rs117308488-cholesterol signal in SAS, Levene's test produces $p=1.6e-15$ in the full sample versus $p=0.48$ when removing minor allele homozygotes. Based on these results, we believe that our original text appropriately emphasizes the need for caution in interpreting the ancestry-specific estimates.

3) The authors applied some methods originally developed for main-effect GWAS test statistics on vQTL test statistics, like multi-ancestry meta-analysis based on Levene's test statistics derived effect size and LDSC. However, the underlying assumptions of these methods may not hold from main effect to variance effect. For example, the derivation/back-transformation from p value and test statistics to effect size and standard error based on Zhu et al. (PMID 27019110, supplementary note 2) was based on simple linear regression, which may be not suitable or to be justified for Levene's test. Note that while the OSCA software provided the effect size and standard error in its output file (<https://yanglab.westlake.edu.cn/software/osca/#vQTLAnalysis>), Wang et al. (PMID: 31453325) did not perform any downstream analysis based on these values. Also did multi-ancestry meta-analysis assume "the effect size is fixed across different ancestry"? How reliable is it for variance or GEI effects? And what LD score dataset should be used in LDSC analysis for multi-ancestry meta-analysis summary data? Did the authors just used that based on European individuals provided by Alkes group (<https://github.com/bulik/ldsc/wiki/LD-Score-Estimation-Tutorial>)?

The reviewer brings up three valuable areas for clarification. Two are addressed briefly immediately below, with the third (appropriateness of beta/SE-based meta-analysis) addressed in greater depth afterwards.

Assumptions of multi-ancestry meta-analysis:

The fixed-effects meta-analysis approach implemented here does assume identical true effect sizes across ancestries. While this may not be strictly true in practice, fixed-effects meta-analysis is common for cross-ancestry GWAS studies (Conti et al. 2021 as a recent example) and we believe that GxE effects should behave similarly. We also acknowledge that random effects may shed light on distinct variance effects across ancestral gradients, and believe this approach should be further explored in future work, possibly in a pooled setting as suggested by Reviewer #1. Finally, we note that we observed limited heterogeneity in vQTL meta-analyses (mean I-squared statistic of 21.0).

Conti DV et al. Trans-ancestry genome-wide association meta-analysis of prostate cancer identifies new susceptibility loci and informs genetic risk prediction. *Nat. Genet.* 2021; 53:65-76.

Ancestry-specificity of LD reference datasets:

We would like to clarify that the genetic correlation analyses were performed using summary statistics from the European-ancestry subset only. We have added the following paragraph to the Methods section with details:

“Genetic correlation analysis was performed using bivariate LD-score regression^{39,40} for both vQTL and ME results from the European-ancestry subset. For each pair of biomarkers, summary statistics were retrieved and analyzed along with a European-ancestry linkage disequilibrium reference dataset from the 1000 Genomes Project (<https://alkesgroup.broadinstitute.org/LDSCORE/>).”

Back-transformed effect estimates and meta-analysis:

As the reviewer notes, OSCA outputs effect sizes based on back-transformation from Levene’s test p-values and effect directions from an additive regression approach. For the specific case of LDSC-based genetic correlation analysis, we believe this is not problematic, because betas are not used quantitatively (only to assign a direction to the z-scores). However, upon further inspection, we agree with the reviewer that quantitative use of the back-transformed betas, as in the current meta-analysis strategy, is inappropriate. So, we have re-run the entire analysis pipeline while changing the meta-analysis strategy to use the p-value-based Stouffer’s method (implemented in METAL as the “SAMPLESIZE” mode). After making this change and re-running the pipeline, the results were not substantially different in a qualitative sense but required systematic updating of affected results throughout the manuscript. Four vQTL loci were gained (across all biomarkers), two were lost, and a subset of the remaining loci had different index variants to be taken into the GEI analysis phase. We refer the reviewers and editors to the tracked-changes version of the manuscript to see all relevant changes, but note that these changes were generally minor and did not change the interpretation of our results.

Minor:

1) More detailed information should be provided for others to understand or replicate this work, which includes:

a. The exact number of (common) genetic variants tested;

Added to Methods section (“UK Biobank genetic data” subsection): “Ultimately, 9,487,345 variants were used for analysis in the European-ancestry subset, with slightly different numbers in the three other ancestry groups based on the MAF filter.”

b. The list of these 2,380 exposures (not only summarised statistics in Table S7);

These exposures are now listed fully in Supp. Table S8. During this process, we noticed that an age variable was included twice based on the output format of the modified PHESANT program used. The impact of this on our results is negligible, but we have included a brief mention of this in the Methods for transparency: “We note that this includes two variables both tracking participant age based on the particulars of the PHESANT functionality.”

c. For the online portal, ancestry-specific summary data, and downloadable text files containing all the variants (not only $p < 0.01$);

These links are now available in the online portal. Due to limitations in data capacity on the CMDKP portal, we are hosting genome-wide summary statistics (vQTL) and all EWIS results (not just $p < 0.01$) for all biomarkers, as suggested by the reviewer, but not ancestry-specific summary data beyond

significant hits as currently provided in a Supplementary Table. We will make these genome-wide, ancestry-specific summary data available on request.

2) Introduction: a few more sentences to introduce the GEI research in the cardiometabolic/cardiovascular field.

We have added the following to the end of the first Introduction paragraph:

“There has been a substantial amount of hypothesis-driven research into GEIs for cardiometabolic traits¹, though these interactions often lack substantial replication^{2,3}. Genome-wide interaction studies have begun to more systematically explore these interactions across the genome⁴, and across multiple exposures⁵, but have focused primarily on body mass index (BMI).”

3) Replication study in WGHS: for 61 vQTLs replicated with a nominal threshold ($p < 0.05$), how many have the same direction?

We chose not to apply directionality to the replication analysis, which was conducted using Levene’s test in R and doesn’t inherently have effect directions. Though OSCA does report back-transformed betas and effect directions, we chose to de-emphasize these directions of effect in general due to the very approximate nature of the regression that assigns these directions in OSCA.

We have clarified this in the Methods section: “Effect directions were not considered when assessing vQTL replication in WGHS.”

4) Two GEI examples in Results section: have “rs139566989” in *LIPC* gene and “rs1229984” in *ADH1B* gene been proved to be the causal variants, while *LIPC* and *ADH1B* are likely to be causal genes due to its relevant biological function? The two SNPs could also be tagging the causal variants through LD.

For *ADH1B*, we do believe that rs1229984 is the causal variant. It is a missense variant in a gene with an extremely strong mechanistic link to alcohol metabolism and there is extensive literature support for this variant and gene impacting both alcohol consumption and alcohol-associated phenotypes. For *LIPC*, rs139566989 is an intronic variant whose only significant eQTL relationship in GTEx is for *LIPC* (in liver and pancreas). It may not be causal; a lipid GWAS fine-mapping analysis by Zubair et al. (citation below) used a trans-ethnic fine-mapping approach to pinpoint rs1077834 (which is in LD with rs139566989: $r^2 = 0.90$ in EUR samples from the 1000 Genomes Project) as the most likely causal variant for the association between *LIPC* and TG. However, these observations and the relevance of the interactions are not dependent on these specific variants being causal.

Zubair N et al., Fine-mapping of lipid regions in global populations discovers ethnic-specific signals and refines previously identified lipid loci, *Human Molecular Genetics*, Volume 25, Issue 24, 15 December 2016, Pages 5500–5512, <https://doi.org/10.1093/hmg/ddw358>.

We have added some of this interesting biological information in the Results text, though we note that the relevance of the interactions is not dependent on causality of the specific variants being tested:

“Though it does not impact the relevance of these findings, we note that rs139566989 may not be causal; a fine-mapping analysis from a trans-ancestry GWAS for lipids identified a nearby variant, rs1077834 ($r^2 = 0.90$ with rs139566989 in the 1000 Genomes Project) as the most likely causal variant for TG in this locus.”

5) 7th paragraph in Discussions (last paragraph in page 7): “they can nonetheless can play a role”. The word “can” appears twice.

We removed the second “can” in that sentence.

6) 2nd paragraph in the “UK Biobank genetic data” subsection of Methods section, it should be “African” instead of “West African”, as I think Kenya (LWK subpopulation) is an East African country.

We originally used the term “West African” to be more specific based on the likely migration and admixture patterns of African-ancestry individuals living in the UK, but we agree that this assumption is not clearly justified in this case and does not match the more diverse African datasets included from the 1000 Genomes Project ancestry reference dataset. We have changed all references to “West African” to simply “African”.

7) The same paragraph: explain more or provide a reference for “probabilistic Gaussian mixture models were built to represent normally-distributed subpopulations ...” and “Rand index score”.

We have updated the text as follows to include more detail and references to our approaches and relevant Python package. We have also included a co-author who led this analysis and was inadvertently left off of the original submission.

“UKB samples were grouped into four primary ancestry groups: African (AFR), East Asian (EAS), European (EUR) and South Asian (SAS). We defined ancestry groups using a Gaussian mixture model on the principal components of 1000 Genomes project (1KGP) phase 3 data. The Gaussian mixture model is a probabilistic model for representing normally distributed subpopulations within an overall population, by assigning data points to the multivariate normal components that maximize the component posterior probability given the data using the expectation-maximization algorithm³⁸. Each component mean was defined by selecting data points randomly without replacement at initialization. We built the Gaussian mixture models using the Scikit-learn Python package and tested 10-fold cross validation on 40 models with different initialization states. We used the adjusted rand index score³⁹ to evaluate the similarity between predicted and true clustering using 1KGP principal components and their 1KGP ancestry labels. This approach was used to refine the subpopulation clustering of 1KGP principal components. We then projected UKB principal components onto the 1KGP principal component space and used the Gaussian mixture model with the highest adjusted Rand index score to cluster individuals into ancestry subgroups.”

8) Figure 4 and 5: indicate the genome-wide or study-wide thresholds. Say, a vertical dash line indicated the genome-wide significant threshold in Figure 5a.

We have added the study-wide significance threshold to Figs. 4a and 4b as a dashed gray line and updated the captions accordingly.

9) The legend of Figure S1, it said the phenotype processing consisted “winsorization”, while in the “serum biomarkers” subsection of Methods section, it said “outliers were set to missing”. I think these two processes are different (<https://en.wikipedia.org/wiki/Winsorizing>).

We thank the reviewer for catching this mistake – we did remove outliers (set to missing) rather than winsorizing. We have changed “winsorization” to “outlier removal” in the Fig. S1 caption.

REVIEWERS' COMMENTS

Reviewer #1 (Remarks to the Author):

This revised manuscript is a substantial improvement on the original submission. The authors addressed many of my questions, with several interesting findings.

1. I wanted to highlight that the authors provided several new examples and analyses that highlight the power of vQTL analysis for discovering interactions. First, within the analysis of non-ME vQTLs, the authors described the rs738408-WC TG interaction, which is a nice example of "sign-flipping" and a locus that would not be discovered in ME loci analysis. Secondly, the contrasting of an EWIS search for GEI using ME loci vs vQTL, while described as "beyond the scope of this manuscript", strongly supports the power of vQTL analysis for GEI discovery in a systematic manner. This analysis is quite large when factoring all the variants, biomarkers, and traits, and (I think) no prior study has done this analysis at comparable scale (or as nicely outlined as the authors do in the reviewer response).

Reviewer Suggestion: I suggest that the authors include further details on this EWIS analysis of ME loci within the final manuscript (or in the supplementary notes, if the authors prefer), as it is a strong addition.

2. The authors also provided a much clearer picture of their results by pinpointing the number of independent GEI, writing:

"Many of the participating exposures were from highly correlated categories such as anthropometric measures. To address this, we iteratively conditioned GEI tests on the most significant exposure interaction at each locus to estimate the number of independent GEIs. We found 66 independent GEIs when retaining the same strict Bonferroni threshold for "secondary signals", which increased to 132 GEIs when using a nominal threshold for these secondary signals."

This reporting is much clearer compared to the original submission reporting only the 846 GEI.

Reviewer Suggestion: For readability, I would add in explicitly what p-values are used within the main text (e.g. $P < 0.05$ for nominal).

3. The authors performed several strong analyses regarding ancestry and meta-analyses in response to both reviewers' comments. However, the authors should explore one final avenue regarding the robustness and benefits of multi-population meta-analysis of vQTLs. The authors write:

"...P-values from the vQTL meta-analysis tracked closely with those from the European subset (Supp. Fig. S2), while uncovering five additional loci that were not significant in the European subset alone."

It appears in Supp Fig S2 that the meta analysis is nearly identical to the European-only analysis. Furthermore, as mentioned in reviewer #2 point #2, the vQTL analysis in the Non-European populations might be particularly sensitive to outliers at low MAF SNPs, and therefore the variance effects are poorly estimated in these populations (with the p-values being unreliable with high false positive rates). This together could suggest that the meta-analysis did not improve upon a European-only analysis. Although the authors disclosed that more additional loci were detected, the authors do not report whether some loci were no longer detected.

I think this leads to an important question to address: does there need to be a "big-enough" sample size for reliable incorporation of different/diverse cohorts in vQTL meta-analysis? If underpowered cohorts are included, does that negatively contribute to a meta-analysis approach? I don't expect the authors to figure out the necessary sample sizes to answer it, but the authors are uniquely positioned to address components of these questions within the realm of their current study.

Currently, the authors only report how many hits were gained through the meta-analysis. This needs 1 additional sensitivity analysis, and a further consideration of the other perspective: that vQTL meta-analysis (with small non-European sample sizes) does not improve upon European-only vQTL analysis.

Reviewer suggestion:

a. For the sensitivity analysis, I suggest that the authors individually inspect the loci which appeared as significant in the meta-analyses but not the European-only analyses. Are these being driven by higher-variance in low-frequency minor-allele homozygotes (within the non-European datasets)? If you remove the homozygotes, do the meta-analysis results still come out as significant?

b. For full clarity, the authors should additionally report how many loci were significant in the European analysis, but not in the meta-analysis, as the way it is currently written addresses the "hits gained" but not the "hits lost". It might be true that the lack of a meta-analysis signal may be due to poor power in the vQTL analysis for diverse populations.

4. On a final note, it is unfortunate to see no epigenomic enrichment of vQTLs compared to ME loci, as that would have been a great discovery (I was hopeful with the large data set that there might be some evidence of functional differences between the two sets). But overall, the authors have nicely added this null analysis within the manuscript, which I still think is a great addition (to show that the analysis was performed, and how it can be performed in similar studies in the future).

Reviewer #2 (Remarks to the Author):

The authors have addressed all my comments.

I would like to correct one mistake in my previous major comment 3, which said "Wang et al. (PMID:31453325) did not perform any downstream analysis based on" back-transformed effect size and standard errors. However, these effect sizes and standard errors were used in the SMR HEIDI and COLOC analysis in Figure S3c by Wang et al.. I apologize for that.

Reviewer #1 (Remarks to the Author):

This revised manuscript is a substantial improvement on the original submission. The authors addressed many of my questions, with several interesting findings.

1. I wanted to highlight that the authors provided several new examples and analyses that highlight the power of vQTL analysis for discovering interactions. First, within the analysis of non-ME vQTLs, the authors described the rs738408-WC TG interaction, which is a nice example of “sign-flipping” and a locus that would not be discovered in ME loci analysis. Secondly, the contrasting of an EWIS search for GEI using ME loci vs vQTL, while described as “beyond the scope of this manuscript”, strongly supports the power of vQTL analysis for GEI discovery in a systematic manner. This analysis is quite large when factoring all the variants, biomarkers, and traits, and (I think) no prior study has done this analysis at comparable scale (or as nicely outlined as the authors do in the reviewer response).

Reviewer Suggestion: I suggest that the authors include further details on this EWIS analysis of ME loci within the final manuscript (or in the supplementary notes, if the authors prefer), as it is a strong addition.

We agree that our large-scale and comprehensive EWIS analysis of ME loci has never been done before at this scale and “highlight[s] the power of vQTL analysis for discovering interactions”. We intend to pursue an in-depth investigation of GEI discovery in ME loci along with a more complete comparative analysis with our vQTL screening approach in a future stand-alone publication, keeping this publication focused on the systematic vQTL screening approach for GEIs.

However, in response to the reviewer’s suggestion, we agree that some of our analyses and results outlined in the reviewer response have never been demonstrated at such scale and should be included in more detail in the manuscript. Given that the EWIS analysis of ME loci is methodologically identical to the EWIS analysis of vQTL loci, we focused on expanding the results in the manuscript as follows:

Original addition:

“We also conducted a full EWIS on the larger set of 4,413 ME-biomarker pairs in order to compare enrichment of GEI findings with those from vQTLs. Replicating findings from previous studies^{15,16}, we saw a substantial enrichment for GEIs within vQTL-biomarker pairs (26.6% of vQTLs had a significant GEI compared to 3.3% for MEs; $p = 1.8 \times 10^{-52}$).”

Detailed addition:

“To understand the power gained for GEI discovery with a vQTL screening approach versus the more conventional approach prioritizing ME loci, we conducted a full EWIS using the same analytical pipeline on the larger set of stage one ME signals and compared GEI enrichment. Replicating findings from previous studies^{15,16} with a much large set of exposures, we found substantial enrichment for GEIs when starting with signals prioritized from vQTLs versus MEs. Specifically, we found that 0.20% (847 / 437,920) of vQTL-biomarker-exposure triplets had a significant GEI versus 0.02% (1,590 / 10,502,940) of ME-biomarker-exposure triplets (Chi-square test $p < 10^{-300}$). This 10-fold discovery enrichment remained after collapsing exposures and comparing variant-biomarker pairs; 26.6% (49 / 184) of vQTL-biomarker pairs had at least one significant GEI, versus just 3.3% (145 / 4,413) of ME-biomarker pairs (Chi-square test $p = 1.8 \times 10^{-52}$). Of note, the EWIS using ME loci used the same Bonferroni-adjusted significance

threshold as the EWIS on vQTL signals; if using MEs in practice, the more stringent Bonferroni adjustment would further reduce the number of GEIs identified.”

2. The authors also provided a much clearer picture of their results by pinpointing the number of independent GEI, writing:

“Many of the participating exposures were from highly correlated categories such as anthropometric measures. To address this, we iteratively conditioned GEI tests on the most significant exposure interaction at each locus to estimate the number of independent GEIs. We found 66 independent GEIs when retaining the same strict Bonferroni threshold for “secondary signals”, which increased to 132 GEIs when using a nominal threshold for these secondary signals.”

This reporting is much clearer compared to the original submission reporting only the 846 GEI.

Reviewer Suggestion: For readability, I would add in explicitly what p-values are used within the main text (e.g. $P < 0.05$ for nominal).

We have updated this sentence as suggested: “We found 66 independent GEIs when retaining the same strict Bonferroni threshold ($p < 2.35 \times 10^{-7}$) for “secondary signals”, which increased to 132 GEIs when using a nominal threshold ($p < 0.05$) for these secondary signals.”

3. The authors performed several strong analyses regarding ancestry and meta-analyses in response to both reviewers’ comments. However, the authors should explore one final avenue regarding the robustness and benefits of multi-population meta-analysis of vQTLs. The authors write:

“...P-values from the vQTL meta-analysis tracked closely with those from the European subset (Supp. Fig. S2), while uncovering five additional loci that were not significant in the European subset alone.”

It appears in Supp Fig S2 that the meta analysis is nearly identical to the European-only analysis. Furthermore, as mentioned in reviewer #2 point #2, the vQTL analysis in the Non-European populations might be particularly sensitive to outliers at low MAF SNPs, and therefore the variance effects are poorly estimated in these populations (with the p-values being unreliable with high false positive rates). This together could suggest that the meta-analysis did not improve upon a European-only analysis. Although the authors disclosed that more additional loci were detected, the authors do not report whether some loci were no longer detected.

I think this leads to an important question to address: does there need to be a “big-enough” sample size for reliable incorporation of different/diverse cohorts in vQTL meta-analysis? If underpowered cohorts are included, does that negatively contribute to a meta-analysis approach? I don’t expect the authors to figure out the necessary sample sizes to answer it, but the authors are uniquely positioned to address components of these questions within the realm of their current study.

Currently, the authors only report how many hits were gained through the meta-analysis. This needs 1 additional sensitivity analysis, and a further consideration of the other perspective: that vQTL meta-analysis (with small non-European sample sizes) does not improve upon European-only vQTL analysis.

Reviewer suggestion:

a. For the sensitivity analysis, I suggest that the authors individually inspect the loci which appeared as significant in the meta-analyses but not the European-only analyses. Are these being driven by higher-variance in low-frequency minor-allele homozygotes (within the non-European datasets)? If you remove the homozygotes, do the meta-analysis results still come out as significant?

First, we note that on further inspection based on this question, we found a mistake in the text: there are ten of these vQTL-biomarker pairs that are significant in the meta-analysis but not the European-only analysis. This is now corrected in the text.

To the reviewer's question about the source of these ten loci: Upon further inspection, they appear to fall into two categories. Seven carry contributions from at least two ancestries with concordant effect directions (five of which include EUR) . The other three appear in the meta-analysis results because they are lower-frequency, ancestry-specific variants (as suggested by the reviewer) that do not appear in other ancestries. These show up because we chose not to require meta-analysis results to carry contributions from multiple ancestries (for example, to allow European-only variants to appear in the final set of results). We have updated the Results text to make this distinction clear:

"P-values from the vQTL meta-analysis tracked closely with those from the European subset (Supp. Fig. S2), while uncovering ten additional loci that were not significant in the European subset alone. Six of these reflected contributions from multiple ancestries, while four were variants that only reached sufficient frequency in one ancestry."

b. For full clarity, the authors should additionally report how many loci were significant in the European analysis, but not in the meta-analysis, as the way it is currently written addresses the "hits gained" but not the "hits lost". It might be true that the lack of a meta-analysis signal may be due to poor power in the vQTL analysis for diverse populations.

We have added the following sentence to that paragraph: "In contrast, nine vQTL-biomarker pairs reached significance in the European-only analysis but not in the meta-analysis."

In total, the Results paragraph discussing ancestry-specificity of vQTL results now reads as follows:

"P-values from the vQTL meta-analysis tracked closely with those from the European subset (Supp. Fig. S2), while uncovering ten additional loci that were not significant in the European subset alone. Six of these reflected contributions from multiple ancestries, while four were variants that only reached sufficient frequency in one ancestry. In contrast, nine vQTL-biomarker pairs reached significance in the European-only analysis but not in the meta-analysis. There were also 71 ancestry-specific vQTLs reaching significance in one or more ancestry-specific analyses but not the meta-analysis (Supp. Table S4), 61 of which were found in non-European ancestry groups. While these ancestry-specific vQTLs may indicate the presence of heterogeneous variance effects across populations due to genetic ancestry differences or ethnic differences in environmental exposures, it is also possible that the non-European findings are the result of spurious associations given lower sample sizes, especially at lower minor allele frequencies¹⁵. Therefore, downstream analysis focuses on the meta-analysis vQTL findings."

4. On a final note, it is unfortunate to see no epigenomic enrichment of vQTLs compared to ME

loci, as that would have been a great discovery (I was hopeful with the large data set that there might be some evidence of functional differences between the two sets). But overall, the authors have nicely added this null analysis within the manuscript, which I still think is a great addition (to show that the analysis was performed, and how it can be performed in similar studies in the future).

We agree with the reviewer and hope to be able to further tease out some of these differences in enrichment in future work with even larger datasets and new phenotypes.

Reviewer #2 (Remarks to the Author):

The authors have addressed all my comments.

I would like to correct one mistake in my previous major comment 3, which said "Wang et al. (PMID:31453325) did not perform any downstream analysis based on" back-transformed effect size and standard errors. However, these effect sizes and standard errors were used in the SMR HEIDI and COLOC analysis in Figure S3c by Wang et al.. I apologize for that.

We again thank the reviewer for their valuable contribution to our manuscript.